# Lamellipodin promotes actin assembly by clustering Ena/VASP proteins and tethering them to actin filaments

**Scott D Hansen[1]\*[†], R Dyche Mullins[1,2]\***

[1]Department of Cellular and Molecular Pharmacology, University of California, San Francisco School of Medicine, San Francisco, United States; [2]Howard Hughes Medical Institute, University of California, San Francisco, United States

**Abstract** Enabled/Vasodilator (Ena/VASP) proteins promote actin filament assembly at multiple locations, including: leading edge membranes, focal adhesions, and the surface of intracellular pathogens. One important Ena/VASP regulator is the mig-10/Lamellipodin/RIAM family of adaptors that promote lamellipod formation in fibroblasts and drive neurite outgrowth and axon guidance in neurons. To better understand how MRL proteins promote actin network formation we studied the interactions between Lamellipodin (Lpd), actin, and VASP, both in vivo and in vitro. We find that Lpd binds directly to actin filaments and that this interaction regulates its subcellular localization and enhances its effect on VASP polymerase activity. We propose that Lpd delivers Ena/VASP proteins to growing barbed ends and increases their polymerase activity by tethering them to filaments. This interaction represents one more pathway by which growing actin filaments produce positive feedback to control localization and activity of proteins that regulate their assembly.

**\*For correspondence:**
sdhansen28@gmail.com (SDH);
Dyche.Mullins@ucsf.edu (RDM)

**Present address:** [†]Quantitative Biology Center, University of California, Berkeley, Berkeley, United States

**Competing interests:** The authors declare that no competing interests exist.

## Introduction

Eukaryotic cells assemble networks of actin filaments adjacent to the plasma membrane to carry out many fundamental processes, including: maintenance of cell morphology; amoeboid locomotion; and cell–cell interaction. The architecture and function of these actin networks is specified in part by the properties of membrane-associated regulatory molecules (*Bear et al., 2002*; *Lacayo et al., 2007*). For example, new filaments are created by nucleation factors recruited to the plasma membrane by Rho-family G-proteins, while fast-growing barbed ends of actin filaments near the membrane interact with regulatory factors that alter the rate and duration of filament elongation. Some factors, including formin-family proteins (*Romero et al., 2004*; *Kovar et al., 2006*) and Enabled/Vasodilator (Ena/VASP stimulated phosphoprotein) proteins (*Barzik et al., 2005*; *Breitsprecher et al., 2008*; *Hansen and Mullins, 2010*; *Breitsprecher et al., 2011*), accelerate filament elongation, while others, such as capping protein, terminate filament elongation (*Dinubile et al., 1995*; *Kuhn and Pollard, 2007*). Local changes in the rates of filament elongation and capping can alter the architecture and function of the actin network and thus tip the balance between membrane protrusion and retraction (*Bear et al., 2002*; *Applewhite et al., 2007*).

One important group of actin regulatory proteins is the Ena/VASP family: weakly processive actin polymerases that accelerate filament elongation and slow filament capping (*Barzik et al., 2005*; *Breitsprecher et al., 2008*; *Hansen and Mullins, 2010*; *Breitsprecher et al., 2011*). Recruitment of Ena/VASP proteins to the plasma membrane often promotes outgrowth of thin, finger-like filopodia and sometimes promotes rapid advance of broad lamellipodial sheets (*Lanier et al., 1999*; *Rottner et al., 1999*). Little is known about the molecular mechanisms that determine localization and activity of Ena/VASP proteins in vivo, but previous work suggests that

**eLife digest** Actin—the most abundant protein in most eukaryotic cells—assembles into a network of filaments that spans the length and breadth of the cell. Like the skeleton of an animal, this 'actin cytoskeleton' gives the cell its shape and strength, and enables the cell to actively move through its environment. To start moving, many cells begin assembling actin filaments next to the cell membrane. The growth of these filaments pushes the membrane forward and creates a two-dimensional structure called a 'lamellipod', which explores the space around the cell and steers its movement.

The actin filaments in a lamellipod are dynamic and undergo repeated cycles of assembly and disassembly. These processes are tightly regulated by a variety of other proteins. Members of the Ena/VASP protein family, for example, collect the building blocks of an actin filament and rapidly stack them in place on the fast-growing end of a filament. The activities of Ena/VASP proteins play an especially important role in creating lamellipodial actin networks and in driving cell movement.

Previous work showed that a protein called Lamellipodin binds to Ena/VASP proteins and helps recruit them to the cell membrane. However, it was unclear whether Lamellipodin could affect the activity of Ena/VASP proteins or their interaction with the actin filaments. Hansen and Mullins have now analyzed the interactions between Ena/VASP, Lamellipodin and actin. The experiments demonstrate that Lamellipodin does not simply tether Ena/VASP proteins to the membrane but also binds directly to actin filaments, via a binding site that is distinct from the site that contacts Ena/VASP. Further experiments with purified proteins revealed that Lamellipodin could interact with both actin filaments and Ena/VASP proteins at the same time. Hansen and Mullins also found that purified Lamellipodin interacted with VASP proteins to form clustered protein complexes, and that together with the tethering of actin filaments to the membrane, this clustering greatly increased VASP's ability to lengthen actin filaments.

By visualizing Lamellipodin tagged with a green fluorescent protein in living cells, Hansen and Mullins then showed that its interaction with actin filaments was sufficient to localize Lamellipodin to the cell membrane. Finally, since Lamellipodin interacts with a multitude of signaling molecules in addition to Ena/VASP proteins, the next big challenge is to understand how Lamellipodin itself is regulated. Future studies could also explore how cells harness the power of the actin cytoskeleton to carry out these essential activities.

a combination of membrane-associated binding partners and free barbed ends work together to recruit and maintain Ena/VASP proteins at the plasma membrane (*Krause et al., 2004*). Stable localization of Ena/VASP to the plasma membrane requires both the Enabled/VASP homology 1 and 2 domains (EVH1 and EVH2) (*Loureiro et al., 2002*; *Applewhite et al., 2007*). The EVH1 domain is a protein–protein interaction motif that binds proline-rich target sequences (e.g., FPPPP) (*Prehoda et al., 1999*), and removal of the EVH1 sequence abolishes localization of Ena/VASP to focal adhesions and the plasma membrane of fibroblasts (*Loureiro et al., 2002*). Interaction with actin filaments is mediated by the EVH2 sequence, which contains an actin binding domain (BD) and a tetramerization motif required for polymerase activity. When expressed by itself in cells, the EVH2 domain concentrates in lamellipodial actin networks, but fails to achieve the same tight, membrane-proximal localization pattern as the intact protein (*Bear et al., 2002*). Cytochalasin D, which caps actin filament barbed ends, rapidly displaces Ena/VASP proteins from the cell periphery (*Krause et al., 2004*; *Lacayo et al., 2007*; *Neel et al., 2009*) arguing strongly that free filament ends play a functional role in Ena/VASP localization.

With the exception of actin, the most well understood Ena/VASP binding partners interact via the EVH1 motif—a globular domain that binds short, proline-rich sequences (*Prehoda et al., 1999*). Ena/VASP binding partners, which include the bacterial effector ActA (*Niebuhr et al., 1997*); the focal adhesion protein, Zyxin (*Drees et al., 2000*); an organizer of dorsal stress fibers, palladin (*Boukhelifa et al., 2004*; *Gateva et al., 2014*); T-cell receptor signaling proteins, Fyb/SLAP (*Krause et al., 2000*); axon guidance factors, Robo/Sax-3 (*Bashaw et al., 2000*); and the *mig-10*/RIAM/Lamellipodin (MRL) proteins (*Krause et al., 2004*; *Lafuente et al., 2004*), all contain tandem repeats of a high-affinity EVH1 binding sequence: D/E FPPPPXD. A second group of Ena/VASP binding partners, including the

formin-family protein Diaphanous (*Grosse et al., 2003*; *Schirenbeck et al., 2006*; *Barzik et al., 2014*; *Bilancia et al., 2014*) and the WAVE regulatory complex (*Law et al., 2013*; *Chen et al., 2014*), relies on a related, but lower affinity, EVH1-binding motif: LPPPPP.

The MRL proteins colocalize with Ena/VASP at the plasma membrane of many different cell types in many diverse organisms, both vertebrates and invertebrates (*Krause et al., 2004*). In the nematode, *Caenorhabditis elegans*, the *mig-10* protein promotes neurite outgrowth and proper axon guidance (*Manser et al., 1997*; *Chang et al., 2006*). Similarly, in mammals the protein Lamellipodin (Lpd) helps determine morphology of neurons and promotes growth of lamellipodial protrusions in a variety of non-neuronal cells (*Michael et al., 2010*; *Pinheiro et al., 2011*; *Yoshinaga et al., 2012*; *Law et al., 2013*). Several MRL proteins, including Lpd, contain a tandem Ras-Association and Pleckstrin Homology (RA-PH) domain which, together with an adjacent coiled-coil region, causes the proteins to form weak homo-dimers in solution (*Chang et al., 2012*). The RA-PH domain is structurally homologous to growth factor receptor-binding proteins Grb7, Grb10, and Grb14 (*Depetris et al., 2009*) but, unlike the Grb proteins, no *direct* interaction between small GTPases (e.g., Ras or Rho family) and MRL proteins has been reported. In vitro, the MRL pleckstrin homology (PH) domain binds phosphatidylinositol lipids: $PI(3,4)P_2$ and $PI(3,4,5)P_2$ and in vivo the PH domain is thought to target these proteins to the plasma membrane in response to extracellular ligands such as PDGF (*Krause et al., 2004*; *Chang et al., 2012*). Although the MRL proteins have been shown to help recruit Ena/VASP proteins to the plasma membrane, their effect on Ena/VASP activity has never been characterized. In addition, results from several studies indicate that the MRL proteins likely have additional, Ena/VASP-independent roles in actin network regulation (*Krause et al., 2004*; *Lyulcheva et al., 2008*; *Michael et al., 2010*).

To better understand the molecular mechanisms underlying cellular control of actin assembly we characterized the interactions between human VASP, Lpd, and filamentous actin in vitro and in live cells. To our surprise, Lpd binds directly to filamentous actin, both in the absence and presence of VASP, an interaction mediated by a cloud of positively charged basic residues scattered through the C-terminal region of the protein (the Lpd Actin Binding Region (ABR), residues 850–1250). In cells, $Lpd^{850-1250aa}$, lacking the RA-PH domain localizes to leading edge membranes and undergoes retrograde flow with the actin cytoskeleton. Surprisingly, the interaction between $Lpd^{850-1250aa}$ and the actin cytoskeleton does not require interactions with Ena/VASP proteins or SH3 containing proteins, such as Abi1 and endophilin. In vitro, Lpd increases processivity of barbed end-associated VASP tetramers by tethering them to actin filaments. Together these results provide mechanistic insight into how growing actin filaments feed back on the polymerases and nucleation promoting factors that regulate their assembly.

## Results

### Lpd binds directly to single actin filaments in vitro

Lpd takes its name from the dynamic lamellipodial actin networks to which it localizes in vivo, even in the absence of Ena/VASP proteins or free actin filament barbed ends (*Krause et al., 2004*). This tenacious localization to leading edge actin networks suggested that Lpd might interact directly with actin filaments, so we tested this idea by simultaneously visualizing monomeric $GFP\text{-}Lpd^{850-1250aa}$ and individual actin filaments in vitro by Total Internal Reflection Fluorescence (TIRF) microscopy (*Figure 1A,B*). In buffer containing 50 mM KCl, the GFP-labeled, monomeric Lpd construct uniformly decorated actin filaments, with a measured dissociation equilibrium constant ($K_d$) of 255 ± 2 nM (*Figure 1B,C*). Consistent with a weak electrostatic interaction, Lpd binding to filamentous actin grew progressively weaker in buffers containing higher concentrations of KCl. In the presence of 100 mM KCl, interaction between monomeric Lpd and single actin filaments were undetectable by TIRF-M (*Figure 1B*). In contrast to these single-filament TIRF assays, we were able to detect interactions between $Lpd^{850-1250aa}$ and filamentous actin by co-sedimentation in the presence of physiological salt concentrations (*Figure 1—figure supplement 1A,B*). The stronger actin filament binding observed by co-sedimentation likely results from Lpd bundling actin filaments in solution.

Conserved acidic residues near the amino terminus of actin create an electronegative patch on the surface of actin filaments that interacts with positively charged, basic residues in many actin binding proteins (*Fujii et al., 2010*). The carboxy-terminal region of human Lpd (residues 850–1250) is highly basic, with an isoelectric point (pI) of 9.97 (*Figure 2A,B*). The distribution of basic residues in this

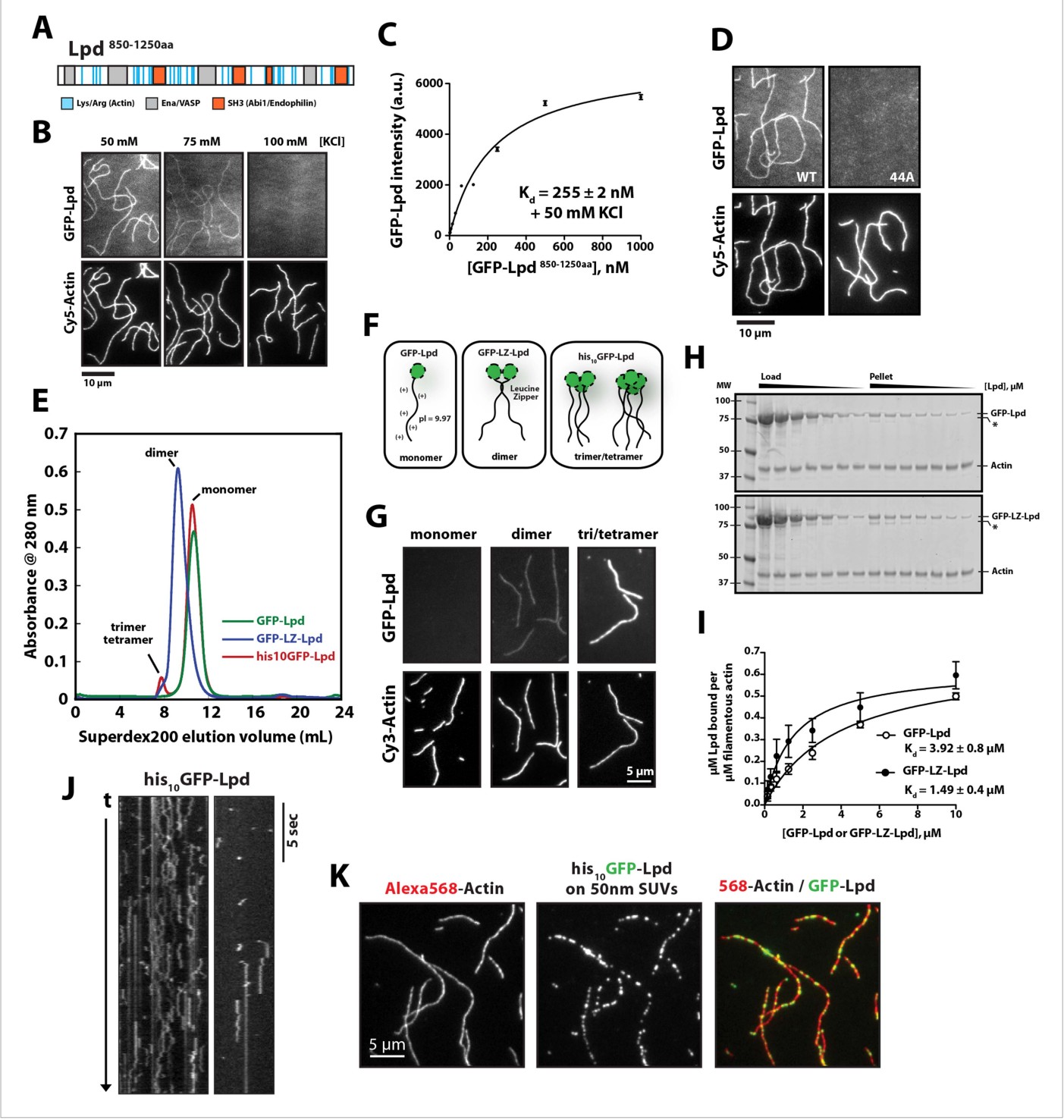

**Figure 1.** Lamellipodin (Lpd) binds directly to single actin filaments in vitro. (**A**) Cartoon representation of the human Lpd[850–1250aa] highlighting the Enabled/Vasodilator (Ena/VASP) binding sites (grey), Abi1/endophilin SH3 binding sites (red), and basic amino acid residues comprising the actin-binding region (blue). (**B**) Representative Total Internal Reflection Fluorescence (TIRF)-M images showing 500 nM monomeric GFP-Lpd[850–1250aa] bound to single actin filaments in the presence of TIRF buffer containing 20 mM HEPES [pH 7.0], 50–100 mM KCl, 1 mg/ml BSA, and 1 mM TCEP. Scale bar, 10 μm. (**C**) Calculation of $K_d$ for GFP-Lpd[850–1250aa] actin filament binding using the average fluorescence intensity of GFP-Lpd bound to phalloidin stabilized actin filaments (20% Cy5 labeled). Error bars represent standard error of the mean. (**D**) Mutations in all 44 lysine/arginine residues to alanine (called Lpd[44A]) abolish F-actin binding of GFP-Lpd[850–1250aa]. Actin filament binding was visualized in the presence 500 nM GFP-Lpd[850–1250aa], wild-type and 44A mutant, in the presence of 50 mM KCl containing buffer as in (**B**). Scale bar, 10 μm. (**E**) Purification of GFP-Lpd[850–1250aa] (monomer), GFP-LZ-Lpd[850–1250aa] (dimer),

*Figure 1. continued on next page*

## Figure 1. Continued

and his$_{10}$-GFP-Lpd$^{850-1250aa}$ (monomers and trimer/tetramer) by size exclusion chromatography. (**F**) Cartoon representation of purified Lpd oligomers in (**E**). (**G**) Oligomerization of GFP-Lpd$^{850-1250aa}$ enhances actin filament binding. Localization of 250 nM GFP-Lpd$^{850-1250aa}$ (monomer), GFP-LZ-Lpd$^{850-1250aa}$ (dimer), and his$_{10}$-GFP-Lpd$^{850-1250aa}$ (oligomers) bound to phalloidin stabilized actin filaments (20% Cy5 labeled) in the presence of TIRF buffer containing 100 mM KCl. Scale bar, 5 µm. (**H**) Representative SDS-PAGE showing co-sedimentation of 1 µM filamentous actin in the presence of increasing concentrations of GFP-Lpd or GFP-LZ-Lpd (0–10 µM monomer concentration). Asterisks (*) on SDS-PAGE gel marks partially translated or proteolyzed GFP-Lpd and GFP-LZ-Lpd that could not be removed during the purification. (**I**) Calculation of K$_d$ for GFP-Lpd and GFP-LZ-Lpd actin binding domains (BDs) by actin co-sedimentation in the presence of 100 mM KCl buffer (± represents error of fit; error bars are S.D. of the mean from two independent experiments). Note that a small fraction of Lpd is non-specifically absorbed to the walls of the centrifuge tubes in the actin co-sedimentation assay. As a result, the stoichiometry of Lpd bound to actin is likely over-estimated by 5–10% (see 'Materials and methods'). (**J**) Kymograph showing diffusion of his$_{10}$-GFP-Lpd$^{850-1250aa}$ oligomers along the length of a phalloidin stabilized actin filaments. Vertical scale bar, 5 s. (**K**) Membrane bound his$_{10}$GFP-Lpd$^{850-1250aa}$ associates with single actin filaments. Localization of 50 nm extruded small unilamellar vesicles (SUVs) DOPC/DOGS-NTA(Ni$^{+2}$) (99:1 molar ratio) coated with his$_{10}$GFP-Lpd$^{850-1250aa}$ bound to Alexa568 phalloidin stabilized actin filaments. 25 nM his$_{10}$GFP-Lpd$^{850-1250aa}$ from (**E**) was combined with 50 nm SUVs (5 µM total lipid containing 1% or 50 nM DOGS-NTA lipid) in buffer containing 20 mM HEPES [pH7], 100 mM KCl, 100 µg/ml BSA, 1 mM TCEP. Scale bar, 5 µm.

The following figure supplement is available for figure 1:

**Figure supplement 1**. Interactions between filamentous actin, GFP-Lpd (850–1250aa), and GFP-LZ-Lpd (850–1250aa) measured by cosedimentation at different buffer ionic strengths.

region is evolutionarily conserved among Lpd homologs, but is not conserved in other Ena/VASP-binding proteins, such as Rap1-GTP-interacting adaptor molecule (RIAM), zyxin, or ActA (*Figure 2B*). Mutating all 44 basic amino acid residues in the C-terminal ABR of Lpd to alanines (to create Lpd (44A)$^{850-1250aa}$) abolished filament binding (*Figure 1D*), but had no effect on the ability of this construct to bind to Ena/VASP proteins (unpublished results). Because the C-terminal region of Lpd appears to be natively unfolded, the loss of actin binding likely reflects simply a loss of electrostatic interactions rather than a change in protein structure. Furthermore, bacterially expressed and purified GFP-Lpd (44A)$^{850-1250aa}$ showed no signs of being structurally or functionally compromised, as compared wild-type GFP-Lpd$^{850-1250aa}$.

Structural studies indicate that full-length (FL) Lpd forms homo-dimers in solution, via both a coiled-coil motif and interactions in the tandem RA-PH domains (*Depetris et al., 2009*; *Chang et al., 2012*). We, therefore, tested the effect of Lpd oligomerization on actin filament binding. We created stable dimers of the C-terminal region of Lpd by fusing Lpd$^{850-1250aa}$ to a dimer-forming, leucine-zipper motif (GFP-LZ-Lpd, *Figure 1E,F*). Unlike the monomeric Lpd constructs, GFP-LZ-Lpd dimers bound to single actin filaments in both low- and high-salt buffers (100 mM KCl; *Figure 1G*). Similarly, dimeric Lpd bound more strongly to filamentous actin, both 'native' and phalloidin stabilized, as compared to monomeric Lpd in a co-sedimentation assay (*Figure 1H,I* and *Figure 1—figure supplement 1*). Based on the ratio of Lpd:Actin sedimented under saturating conditions, we estimate a stoichiometry of one GFP-Lpd$^{850-1250aa}$ to at least two actin protomers.

We also purified spontaneously formed trimeric and tetrameric oligomers of his$_{10}$GFP-Lpd$^{850-1250aa}$ by size-exclusion chromatography and found that their affinity for filamentous actin was even further enhanced (*Figure 1E,F*). In our TIRF assay we observed oligomeric his$_{10}$GFP-Lpd$^{850-1250aa}$ particles bind and diffuse linearly along the sides of actin filaments (*Figure 1J*), an activity we previously observed for tetrameric Cy3-VASP constructs (*Hansen and Mullins, 2010*). Finally, we observed enhanced filament binding when we coupled freshly gel-filtered, monomeric his$_{10}$GFP-Lpd$^{850-1250aa}$ to small unilamellar vesicles (SUVs) containing DOGS-NiNTA lipids (*Figure 1K*). These Lpd-coated SUVs bound tightly to individual actin filaments in vitro, demonstrating that oligomerization and/or clustering of Lpd$^{850-1250aa}$ promotes stable actin filament binding, even in near physiological salt concentrations.

## Membrane-tethered Lpd slows dendritic actin network assembly in vitro

We next tested whether Lpd can interact directly with artificial lamellipodial actin networks reconstituted in vitro from purified components (*Loisel et al., 1999*; *Akin and Mullins, 2008*). We used the Arp2/3 complex, together with capping protein, to assemble dendritic actin networks on lipid-coated bead (LCBs) containing Ni-conjugated lipids bound to his$_{10}$Cherry-SCAR. In these assays

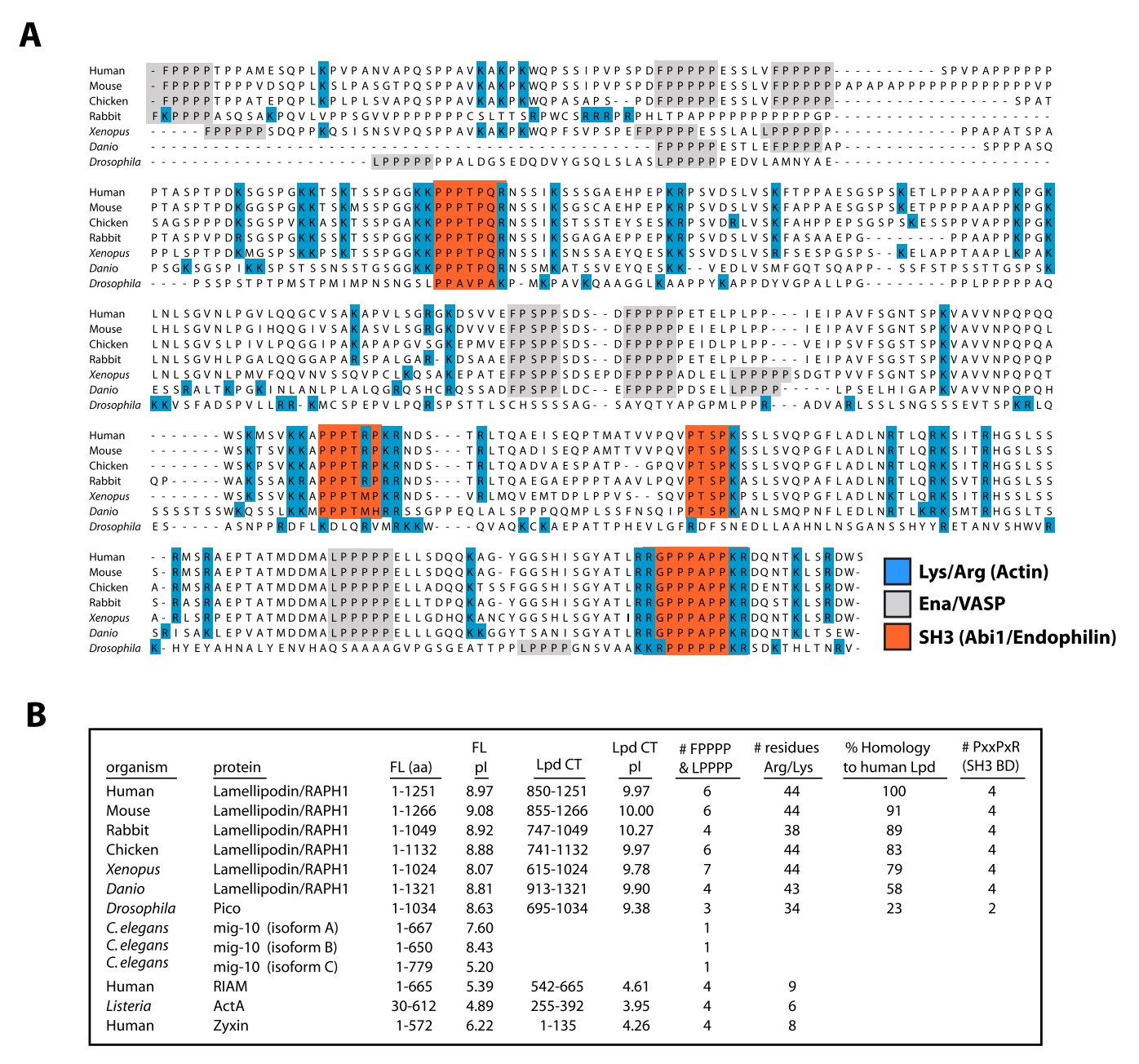

**Figure 2**. Conservation of Lpd (850–1250aa) amino acid sequence and isoelectric point (pI). (**A**) Protein sequence alignment of human Lpd and homologs C-termini. Basic amino acid residues (arginine and lysine) are highlighted in blue. Gray boxes mark the location of the canonical Ena/VASP homology 1 (EVH1) BDs (i.e., FPPPP or LPPPP), while red boxes highlight the predicted Abi1/endophilin SH3 domain binding sites (i.e., PxxPxR). Secondary structure prediction algorithms suggest that the Lpd (850–1250aa) lacks secondary structure (data not shown). (**B**) Comparison of Lpd, Pico, *mig-10*, RIAM, ActA, and Zyxin pIs across the canonical Ena/VASP BD containing one or more FPPPP motifs. Domain boundaries for this region are termed, Lpd C-terminus (Lpd CT). The number of arginine and lysine residues were calculated across the region specified 'Lpd C-terminus (Lpd CT)'. The number of Ena/VASP and SH3 domain binding sites were counted across the domain boundaries defined, Lpd CT, and are shown in columns five and six, respectively. The pIs for Lpd CT were calculated using EXPASY (*Wilkins et al., 1999*). Abbreviations are as follows: full-length (FL); binding-domain (BD); Lamellipodin (Lpd).

we also included either membrane tethered his$_{10}$GFP or his$_{10}$GFP-Lpd$^{850-1250aa}$ (*Figure 3A*, *Figure 3—figure supplement 1*). Since his$_{10}$GFP-Lpd$^{850-1250aa}$ is tethered to the membrane via Ni-conjugated lipids, our reconstitution lacks the recruitment and dissociation dynamics that may be

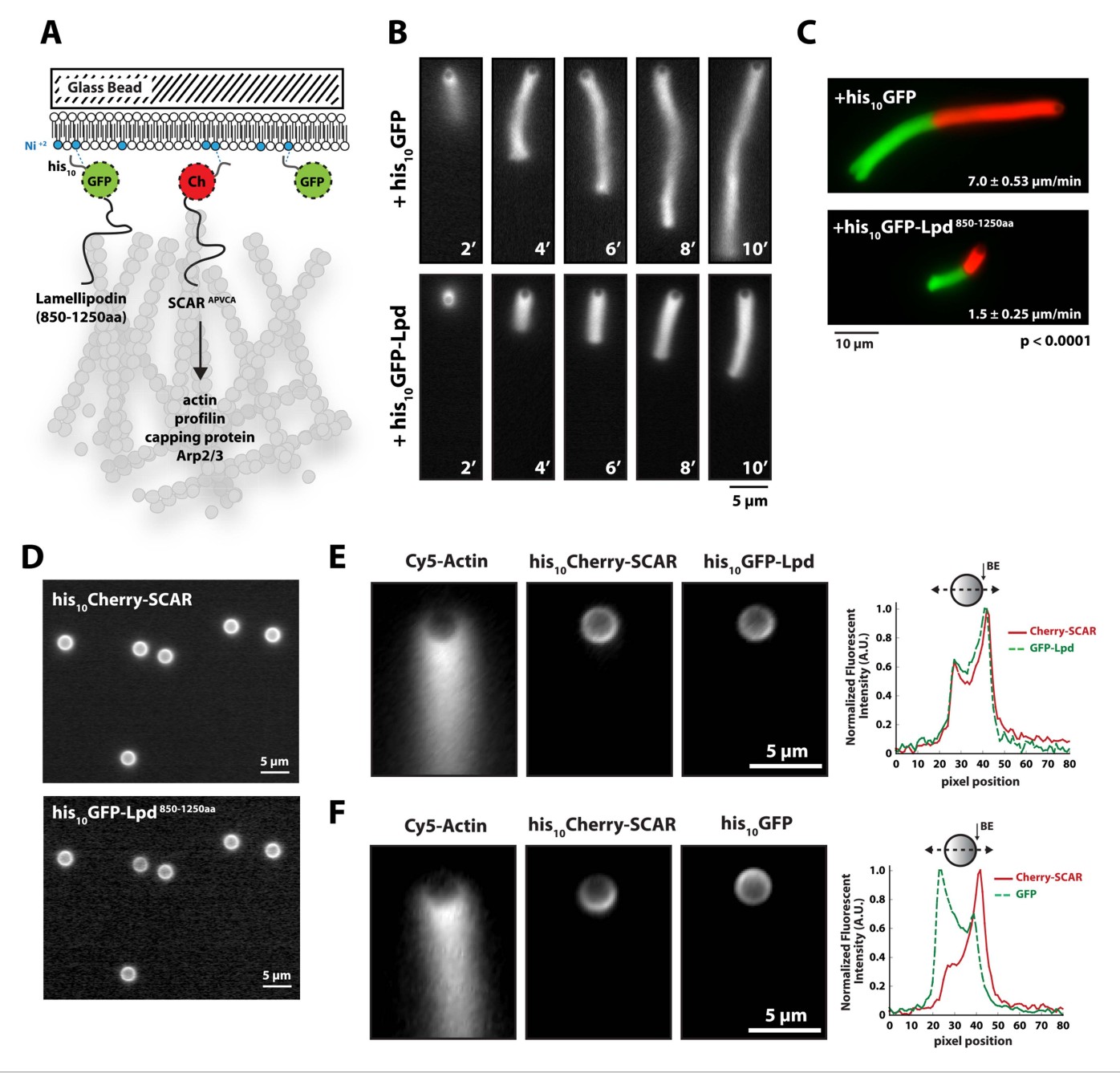

**Figure 3**. Membrane-tethered Lpd slows dendritic actin network assembly in vitro. (**A**) Cartoon illustrating his$_{10}$Cherry-SCAR$^{APWCA}$, his$_{10}$GFP-Lpd$^{850-1250aa}$, and his$_{10}$GFP tethered to a lipid coated beads (LCBs) containing DOGS-NTA(Ni) lipid (blue head groups). Actin network assembly on the bead surface is initiated by adding monomeric actin, profilin 1, Arp2/3, capping protein, and buffer containing KCl. (**B**, **C**) Membrane tethered his$_{10}$-GFP-Lpd$^{850-1250aa}$ slows actin network assembly on LCBs. (**B**) Representative actin comet tails assembled in the presence of 7.5 μM actin (5% Alexa488-Actin), 3 μM hProfilin 1, 100 nM Arp2/3, 100 nM capping protein, and buffer containing 150 mM NaCl. LCBs (2.3 μm, 4% DOGS-NTA(Ni): 96% DOPC) were charged with 75 nM his$_{10}$-Cherry-SCAR$^{APWCA}$, plus 25 nM his$_{10}$-GFP-Lpd$^{850-1250aa}$ or 25 nM his$_{10}$-GFP (i.e., 75% his$_{10}$-Cherry, 25% his$_{10}$-GFP). Actin network assembly and disassembly was stopped at the indicated time points by combining the bead motility assay, 1:1, with 37.5 μM Latrunculin B-phalloidin mixture. Scale bar, 5 μm. (**C**) Representative actin comet tails assembled as in (**B**) for 5 min before transitioning from actin motility mix with 7.5 μM actin (5% Alexa488 labeled, GREEN) into an identical mix, but containing 7.5 μM actin (5% Cy3-Actin, RED). The length of Cy3-actin incorporated into the comet tail was measured to determine the growth velocity of multiple tails (n ≥ 50 tails). Error (±) represents the standard deviation of the mean (p-value = 3 × 10$^{-29}$; two-tailed t-test for data sets with equal variance). Scale bar, 10 μm. (**D**) Homogenous distribution of his$_{10}$-Cherry-SCAR$^{APWCA}$ and his$_{10}$GFP-Lpd$^{850-1250aa}$ before initiating actin network assembly. Scale bar, 5 μm. (**E**, **F**) Spatial distribution of his$_{10}$Cherry-SCAR$^{APWCA}$, his$_{10}$GFP-Lpd$^{850-1250aa}$, and his$_{10}$GFP during steady state actin tail growth and recycling (30 min time point). Actin networks were assembled in the presence of 7.5 μM actin (5% Alexa488), 3 μM hProfilin 1, 100 nM

*Figure 3. continued on next page*

**Figure 3. Continued**

Arp2/3, 100 nM Mm capping protein, and 3 µM hCofilin. (**E**) his$_{10}$Cherry-SCAR$^{APWCA}$ and his$_{10}$GFP-Lpd$^{850-1250aa}$ concentrate on the barbed end dense side of the actin comet tail. (**F**) his$_{10}$-Cherry-SCAR$^{APWCA}$ concentrates on the barbed end dense side of the actin comet tail, while his$_{10}$-GFP is excluded from the barbed end attachment zone. Line scans across LCBs are shown to the right. Scale bar, 5 µm.

The following figure supplement is available for figure 3:

**Figure supplement 1**. Actin based motility on lipid coated glass beads.

mediated by interactions between the Lpd PH domain and phosphatidylinositol lipids (i.e., PI(3,4)P$_2$). Nonetheless, we used this assay to test whether membrane tethered Lpd modulates actin network assembly in an autonomous manner. By measuring the length and density of actin comet tails at various times (*Figure 3B*), we found that membrane-tethered Lpd significantly slowed the rate of actin network assembly (1.5 ± 0.25 µm/min) compared to membrane-tethered GFP alone (7.0 ± 0.48 µm/min; *Figure 3B,C*).

In addition to slowing network growth, membrane-tethered Lpd also polarized in an actin-dependent manner on the surface of LCBs. Similar to the polarization of N-WASP on lipid vesicles in *Xenopus* egg extract (*Co et al., 2007*), we found that Scar/WAVE and Lpd constructs become enriched in regions of the membrane most closely associated with the growing actin network. In the absence of actin, both his$_{10}$Cherry-SCAR and his$_{10}$GFP-Lpd$^{850-1250aa}$ were distributed uniformly around the LCBs (*Figure 3D*). Within 10 min of initiating actin network assembly on LCBs, however, both membrane-tethered Scar/WAVE and Lpd constructs (his$_{10}$GFP-Lpd$^{850-1250aa}$) became concentrated in the region adjacent to the actin network, a region we refer to as the 'barbed end attachment zone' (*Figure 3E*, *Figure 3—figure supplement 1B–D*). In contrast, membrane-tethered his$_{10}$GFP was displaced *away* from the barbed end attachment zone and became concentrated on the opposite side of the microsphere (*Figure 3F*).

## Lpd interacts with the actin cytoskeleton in vivo

Having found that Lpd binds actin filaments in vitro, we next worked to determine whether actin binding by Lpd plays a significant role in its cellular localization and function. To address this question we visualized the localization of fluorescently tagged, FL Lpd (GFP-Lpd$^{1-1250aa}$) in *Xenopus* Tissue Culture (XTC) cells, spread on poly-L-lysine (PLL)-coated coverslips (*Figure 4A*). We used XTC cells because they spread well and produce very thin peripheral actin networks, well suited for fluorescence microscopy. Ectopic gene expression from a truncated cytomegalovirus (CMV) promoter produces very low levels of protein expression in XTC cells, ideal for single-molecule fluorescence microscopy (*Watanabe and Mitchison, 2002*). TIRF microscopy imaging of single GFP-Lpd$^{1-1250aa}$ molecules revealed that FL Lpd localizes predominantly to the leading edge of ruffling membranes and cycles on and off the plasma membrane on a subsecond time scale (*Figure 4B*, *Figure 4—figure supplement 1*, *Video 1*). Leading edge membrane localization of GFP-Lpd$^{1-1250aa}$ required dynamic actin assembly and disassembly, because acute treatment of cells with a chemical inhibitor cocktail that freezes actin dynamics (JLY drug cocktail: Jasplakinolide, Latrunculin B, and Y27632 Rock kinase inhibitor [*Peng et al., 2011*]) caused rapid loss of membrane-localized Lpd (*Figure 4C*).

To determine which Lpd binding partners—actin, Ena/VASP, Abi1/endophilin, inositol phospholipids, or Ras/Rho-family G-proteins—are required for leading edge localization, we compared the localization of various Lpd truncation mutants in XTC cells. We discovered that not only does GFP-Lpd$^{850-1250aa}$, which lacks the tandem RA-PH domains, localize to leading edge membranes in XTC cells (*Figure 4C*), this construct also undergoes retrograde flow with the lamellipodial actin network. Analysis of speckle velocities in XTC cells co-expressing GFP-Lpd$^{850-1250aa}$ and mCherry-Actin revealed that the two proteins move with the same mean velocity: 71.9 ± 17.5 nm/s for Lpd$^{850-1250aa}$ vs 73.5 ± 14 nm/s for actin (*Figure 4F*, *Video 2*). In contrast to mCherry-Actin, GFP-Lpd$^{850-1250aa}$ speckles were observed less frequently, appeared more diffuse, and had shorter lifetimes (*Figure 4—figure supplement 2*). We suspect that the larger GFP-Lpd$^{850-1250aa}$ fluorescent particles are associated with fast endophilin-mediated endocytosis (FEME, *Vehlow et al., 2013*; *Boucrot et al., 2015*). Consistent with constitutively dimeric Lpd having a higher affinity for actin, GFP-LZ-Lpd$^{850-1250aa}$ displayed slightly

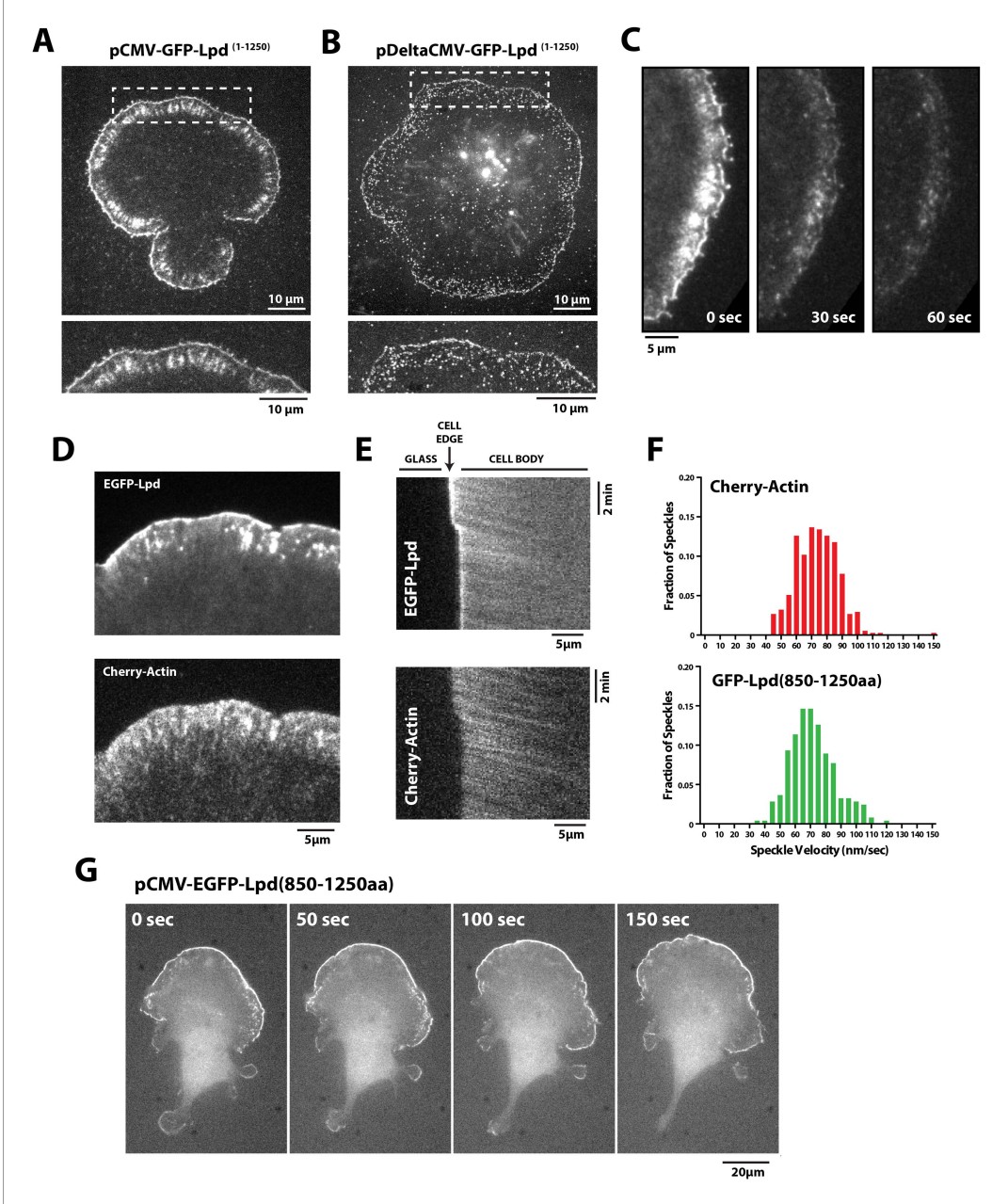

**Figure 4**. Lpd (850–1250aa) localizes to the leading edge membranes and undergoes retrograde flow with the actin cytoskeleton. (**A**, **B**) Plasma membrane localization of GFP-Lpd[1–1250aa] visualized with TIRF microscopy in Xenopus Tissue Culture (XTC) cells spread on poly-L-lysine (PLL). GFP-Lpd[1–1250aa] was ectopically expressed from a (**A**) cytomegalovirus (CMV) or (**B**) DeltaCMV promoter. (**B**) Maximum intensity projection of a XTC cell expressing a single molecule concentration of GFP-Lpd[1–1250aa]. Scale bar, 10 µm. Leading edge membrane marked by dashed box is enlarged below. Scale bar, 5 µm. (**C**) Leading edge membrane localization of GFP-Lpd[1–1250aa] in XTC cell viewed by TIRF-M following the addition of 8 µM Jasplakinolide, 10 µM Latrunculin B, and 10 µM Y27632 (Rock kinase inhibitor) (*Peng et al., 2011*). Scale bar, 5 µm. (**D**) Representative image of XTC cell coexpressing GFP-Lpd[850–1250aa] and mCherry-Actin. Scale bar, 5 µm. (**E**) Kymographs show retrograde flow of GFP-Lpd[850–1250aa] and mCherry-Actin. Scale bar, 5 µm. (**F**) Histogram showing distribution of GFP-Lpd[850–1250aa] and mCherry-Actin speckle velocities. Mean speckle velocities of 71.9 ± 17.5 nm/s (n = 246 speckles) and 73.5 ± 14 nm/s (n = 373 speckles) were calculated for GFP-Lpd[850–1250aa] and mCherry-Actin, respectively. (**G**) GFP-Lpd[850–1250aa] localizes to the leading edge membrane of polarized mouse B16F1 cell migrating on laminin coated glass substrate. Scale bar, 20 µm.

*Figure 4. continued on next page*

*Figure 4. Continued*

The following figure supplements are available for figure 4:

**Figure supplement 1**. Localization of GFP-Lpd (850–1250aa) and GFP-LZ-Lpd (850–1250aa).

**Figure supplement 2**. Retrograde flow of GFP-Lpd (850–1250aa) and GFP-LZ-Lpd (850–1250aa) with the actin cytoskeleton.

more robust leading edge membrane localization and the intensity of presumptive endocytic particles were brighter, as compared to those observed in cells expressing GFP-Lpd$^{850-1250aa}$ (*Figure 4—figure supplements 1, 2*).

We also transiently expressed GFP-Lpd$^{850-1250aa}$ in polarized B16F1 mouse melanoma cells migrating on laminin coated glass, and observed the same leading edge membrane localization as in XTC cells (*Figure 4G*, *Figure 4—figure supplement 1*, *Video 3*). Importantly, this localization does not require acute receptor stimulation following serum starvation, implying that membrane recruitment of GFP-Lpd$^{850-1250aa}$ represents the default Lpd localization in these migrating cells.

We next asked whether GFP-Lpd$^{850-1250aa}$ requires Ena/VASP binding for leading edge membrane localization in XTC cells. The C-terminal region of Lpd contains six FPPPP peptide sequences, each of which can potentially bind one VASP EVH1 domain. By analytical ultracentrifugation, we found that GFP-Lpd$^{850-1250aa}$ simultaneously binds up to four VASP EVH1 domains in solution (*Figure 5—figure supplement 1*). We mutated all six FPPPP motifs in Lpd$^{850-1250aa}$ to AAPPP (*Figure 5—figure supplement 2*) and expressed this mutant protein as either a his$_{10}$-GFP fusion in *Escherichia coli* (his$_{10}$-GFP-Lpd(AAPPPx6)$^{850-1250aa}$) or a GFP fusion in XTC cells (GFP-Lpd(AAPPPx6)$^{850-1250aa}$). Although the his$_{10}$-GFP-Lpd(AAPPPx6)$^{850-1250aa}$ mutant failed to recruit Ena/VASP proteins to LCBs in vitro (*Figure 5A*), the localization of the GFP-Lpd(AAPPPx6)$^{850-1250aa}$ mutant in XTC cells was nearly indistinguishable from GFP-Lpd$^{850-1250aa}$ (*Figure 5B*). The only noticeable difference was a slight decrease in the amount of protein associated with nascent focal adhesions (*Figure 5B*).

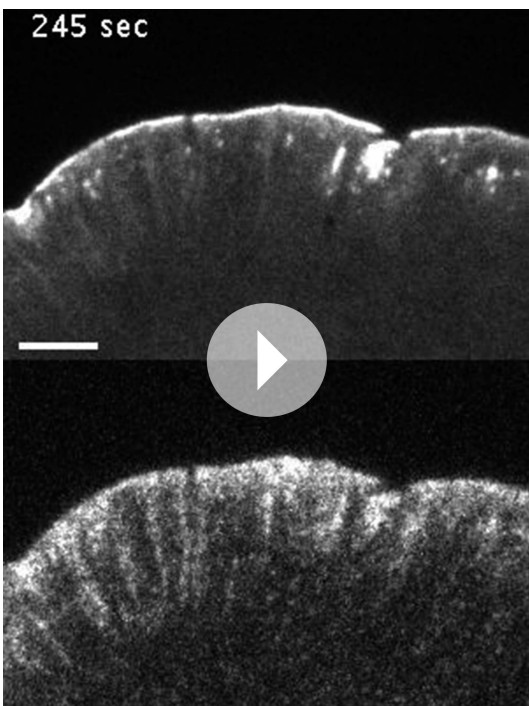

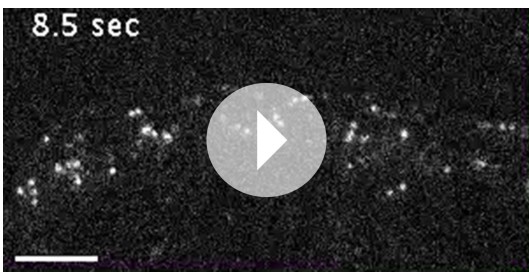

**Video 1.** Localization of full length GFP-Lpd (1 –1250aa) expressed at single molecule concentrations from the DeltaCMV promoter in *Xenopus* Tissue Culture (XTC) cells. Images were acquired with temporal resolution of 100 ms using Total Internal Reflection Fluorescence (TIRF)-M. Video plays at 15 frames per second. Scale bar, 5 μm.

**Video 2.** Retrograde flow of GFP-Lpd$^{850-1250aa}$ and mCherry-Actin in XTC cells spread on poly-L-lysine (PLL) coated coverslips. Cell was imaged with a temporal resolution of 5 s using TIRF-M at 23°C. GFP-Lpd$^{850-1250aa}$ and mCherry-Actin were ectopically expressed from either a CMV or DeltaCMV promoter, respectively. Video plays at 10 frames per second. Scale bar, 5 μm.

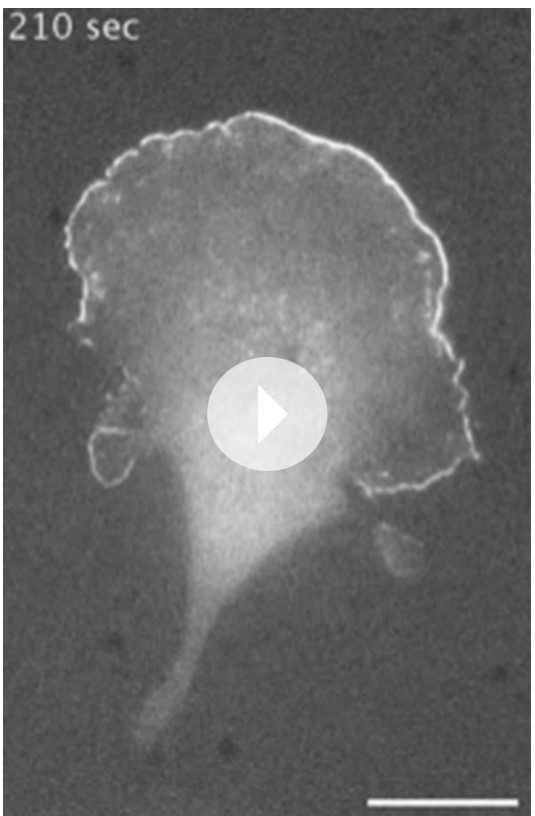

**Video 3.** Leading edge membrane localization of GFP-Lpd$^{850-1250aa}$ in polarized B16F1 mouse melanoma cell migrating on laminin coated glass. B16F1 cell was imaged with a temporal resolution of 10 s using wide-field epi fluorescence at 37˚C. GFP-Lpd$^{850-1250aa}$ was ectopically expressed from a CMV promoter. Video plays at 10 frames per second. Scale bar, 20 µm.

FL human Lpd$^{1-1250aa}$ contains 10 potential SH3 domain binding sites (*Law et al., 2013*), of which four fall within our ABR construct, Lpd$^{850-1250aa}$ (*Figure 2A*). To abolish the known interactions with Abi1 and endophilin, we mutated essential proline residues (e.g., PxxPxR → GxxGxR) in all four SH3 domain binding sites (*Figure 5—figure supplement 2*). This mutant, termed Lpd(SH3*)$^{850-1250aa}$, could still localize to the leading edge in XTC cells (*Figure 5B*), but to a lesser extent than wild-type Lpd or the (AAPPP)$_{x6}$ Lpd mutant which does not bind Ena/VASP EVH1 domains. Combining the (AAPPP)$_{x6}$ and SH3* mutations in the same protein further reduced the percentage of cells with leading edge membrane localization of these GFP-Lpd constructs (*Figure 5B*). In each case, mutations in the Ena/VASP or Abi1/endophilin binding sites resulted in a loss of leading edge localization that coincided with an increased cytoplasmic localization of the GFP-Lpd$^{850-1250aa}$ construct.

To test whether the cloud of basic residues in the C-terminal region of Lpd mediates interactions with the actin cytoskeleton in vivo, we created a mutant lacking lysine and arginine residues that flank the multiple Ena/VASP and Abi1/endophilin binding sites (*Figure 5—figure supplement 2*). In vitro, we found that GFP-Lpd (44A)$^{850-1250aa}$ lacking all lysine and arginine residues failed to bind to filamentous actin in vitro (*Figure 1D*). Because several of these point mutations flank or fall within the SH3 binding sites, we also generated a GFP-Lpd(35A)$^{850-1250aa}$ actin binding mutant, which lacks only the basic residues that sit between Ena/VASP and Abi1/endophilin binding sites (*Figure 5—figure supplement 2*). Similar to GFP-Lpd(44A)$^{850-1250aa}$, the GFP-Lpd(35A)$^{850-1250aa}$ construct also failed to concentrate at actin-based membrane protrusion or nascent focal adhesions in XTC cells (*Figure 5B*). We attempted to increase the avidity of GFP-Lpd(35A)$^{850-1250aa}$ for the leading edge membrane by expressing the protein as a leucine zipper-mediated constitutive dimer (GFP-LZ-Lpd (35A)$^{850-1250aa}$), but this mutant remained cytoplasmic (*Figure 5B*).

To determine the role of membrane association in the leading-edge localization of Lpd we attempted to rescue leading edge localization of various GFP-Lpd$^{850-1250aa}$ mutants by constitutively tethering each protein to the plasma membrane with a Lyn palmitolyation sequence derived from Src kinase (*Figure 5C*) (*Inoue et al., 2005*). Similar to what has previously been observed for membrane tethered ActA-(FPPPP)$_{x4}$-CAAX (*Bear et al., 2002*) and soluble GFP-Lpd$^{850-1250aa}$, described here, Lyn-GFP-Lpd$^{850-1250aa}$ localized to the leading edge of actin-based membrane protrusions (*Figure 5D*, *Video 4*). In contrast, leading edge localization of the combined (AAPPP)$_{x6}$ and SH3* mutant construct was observed in fewer cells than the Lyn-GFP-Lpd$^{850-1250aa}$. Although Lyn-GFP-Lpd(35A)$^{850-1250aa}$ was targeted to the plasma membrane by the palmitoyl lipid anchor, this mutant still failed to localize to the peripheral regions where Ena/VASP, Abi1, and endophilin are located (*Figure 5D*). Localization of Lyn-GFP-Lpd(35A)$^{850-1250aa}$ was identical to that of a membrane-anchored Lyn-GFP construct lacking any Lpd sequences. Both of these constructs were distributed uniformly across the plasma membrane (*Figure 5D*).

Given this loss of positive charges, it is possible that failure of Lyn-GFP-Lpd (35A) to localize to the leading edge membrane reflects an inability to interact with a polarized population of anionic lipids.

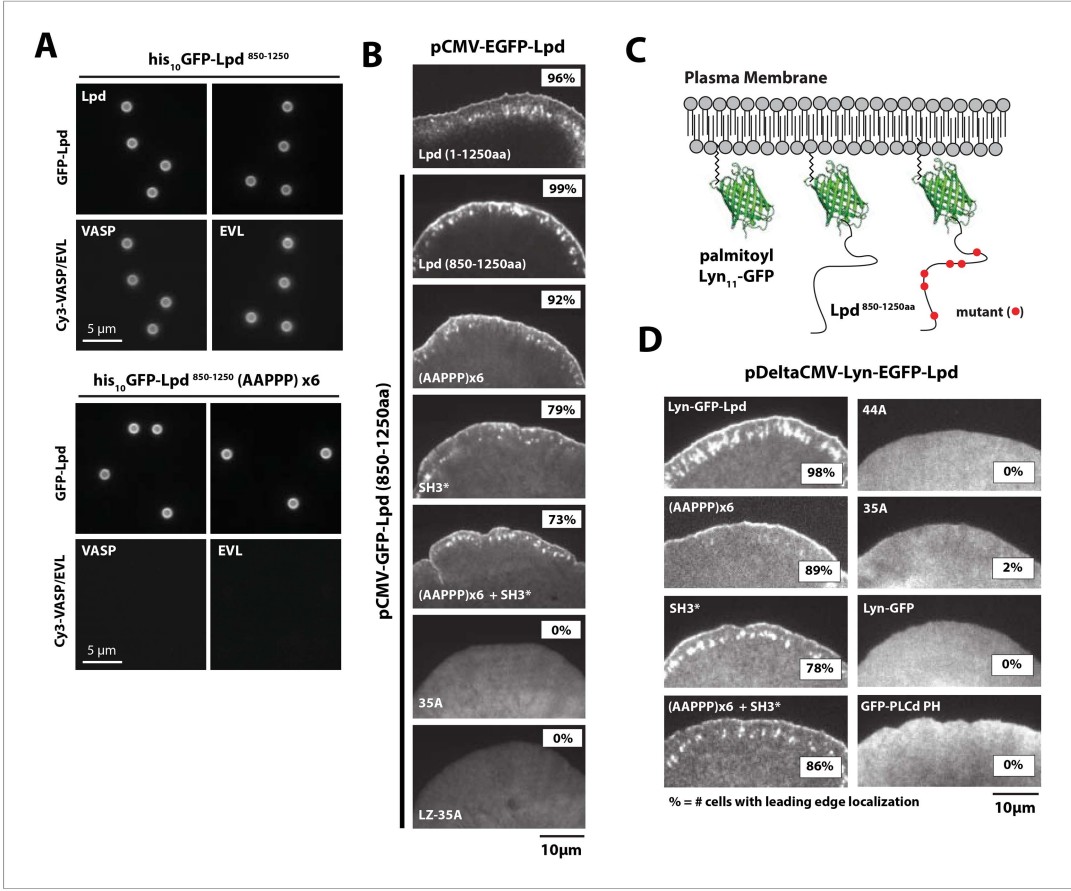

**Figure 5**. Interactions with Ena/VASP or Abi1/endophilin are not required for Lpd (850–1250aa) membrane localization. (**A**) Lpd FPPPP peptide sequences are required to recruit Ena/VASP proteins to the lipid coated beads (LCBs). Glass microspheres were coated with SUVs containing DOPC/DOGS Ni-NTA lipids (96:4 molar ratio). LCBs were then incubated with 100 nM his10-GFP-Lpd$^{850–1250aa}$, (wild-type and AAPPPx6 mutants) for 15 min, before being mixed with 500 nM Cy3-VASP or Cy3-EVL. Lpd mutant, (AAPPP)x6, cannot recruit purified Cy3-VASP or Cy3-EVL to LCBs. (**B**) Basic residues in flanking the Ena/VASP and Abi1/endophilin binding sites are required for leading edge localization of GFP-Lpd$^{850–1250aa}$ in XTC cells. Representative images of wild-type and mutant GFP-Lpd$^{850–1250aa}$ protein in XTC cells. Localization of full length GFP-Lpd$^{1–1250aa}$ (top panel) is shown for comparison. Scale bar, 5 μm. Refer to *Figure 5—figure supplement 2* for amino acid sequences of each Lpd$^{850–1250aa}$ mutant. (**C**) Cartoon schematic showing palmitoylated Lyn-GFP and Lyn-GFP-Lpd (WT and FPPPP → AAPPP$_{x6}$ mutant) anchored in the plasma membrane. The crystal structure of GFP was derived from *Yang et al. (1996)* (1GFL.pdb). (**D**) Constitutively membrane tethered Lyn-GFP-Lpd$^{850–1250aa}$ localizes to the leading edge. Leading edge localization of Lyn-GFP-Lpd$^{850–1250aa}$ does not require interactions with Ena/VASP proteins or Abi1/endophilin. Localization of Lyn-GFP and GFP-PLCδ (pleckstrin homology [PH] domain that binds to PI(4,5)P$_2$), phenocopied the uniform membrane localization of Lyn-GFP-Lpd (35A and 44A). Scale bar, 10 μm. (**B**, **D**) The percentage of cells with leading edge localization is indicated in the upper right-hand corner of each representative image (n = 96–167 cells imaged for each GFP-Lpd$^{850–1250aa}$ construct expressed in XTC cells).

The following figure supplements are available for figure 5:

**Figure supplement 1**. Lpd-VASP binding stoichiometry determined by sedimentation equilibrium.

**Figure supplement 2**. Lpd (850–1250aa) wild-type and mutant protein sequence alignment.

**Figure supplement 3**. Membrane tethered Lyn-GFP-Lpd (850–1250aa) requires basic residue for leading edge localization.

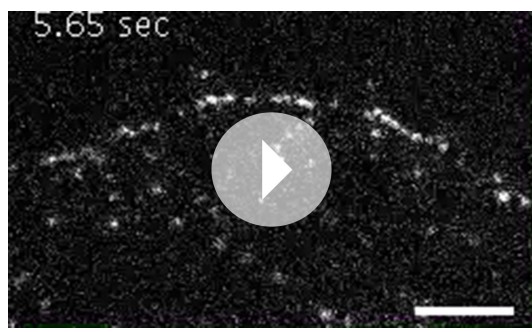

**Video 4.** Localization of membrane tethered Lyn-GFP-Lpd (850–1250aa) at the leading edge membrane of XTC cells spread on PLL coated coverslips expressed at single molecule concentrations from DeltaCMV promoter. Images were acquired using TIRF microscopy with temporal resolution of 50 ms time. Video plays at 15 frames per second. Scale bar, 5 μm.

To look for such a polarized lipid distribution we expressed a phosphatidylinositol lipid sensor, based on the pleckstrin-homology domain of phospholipase Cδ (GFP-PLCδ PH). We found that GFP-PLCδ PH distributes uniformly across the plasma membrane of XTC cells (*Figure 5D*). This result supports our interpretation that failure of Lyn-GFP-Lpd (35A) to localize to leading edge membranes in XTC cells reflects the fact that Lpd coupling to the actin cytoskeleton is required for establishing interactions with Ena/VASP or Abi1/endophilin at actin based membrane protrusions.

To test whether leading edge membrane localization of GFP-Lpd$^{850–1250aa}$ requires inter-action with free actin filament barbed ends or with the sides of actin filaments, we used pharmacological perturbations to examine how GFP-Lpd$^{850–1250aa}$ leading edge localization responds to changes in actin cytoskeletal dynamics. First, we simply froze actin assembly and disassembly by adding a mixture of Jasplakinolide and Latrunculin B to cells, without inhibiting myosin activity. Under these conditions we observed rapid retrograde movement of GFP-Lpd$^{850–1250aa}$ away from the leading edge, identical to the movement of mCherry-Actin (*Figure 6A*, *Video 5*). The dynamics of GFP-Lpd$^{850–1250aa}$ translocation suggests that, in the absence of net actin assembly, myosin can pull GFP-Lpd$^{850–1250aa}$ off the leading edge via a tight linkage to the actin cytoskeleton. In these experiments, a fraction of membrane-localized GFP-Lpd$^{850–1250aa}$ remained at the leading edge, presumably due to the maintenance of interaction with Ena/VASP and/or Abi1/endophilin.

Next we treated cells expressing GFP-Lpd$^{850–1250aa}$ with Cytochalasin D, a small molecule that caps barbed ends of actin filaments. Consistent with previous observations (*Krause et al., 2004*; *Neel et al., 2009*), addition of Cytochalasin D to XTC cells strongly inhibited the localization of GFP-VASP to leading edge membranes (unpublished observations). When we treated these cells with a 100 nM Cytochalasin D, we observed a large fraction of leading edge-localized GFP-Lpd$^{850–1250aa}$ immediately dissociate from the leading edge membrane and flow back with the actin cytoskeleton (*Figure 6B*, *Video 6*). Because GFP-LZ-Lpd$^{850–1250aa}$ binds more strongly to filamentous actin, dissociation from the leading edge membrane was more dramatic following addition of Cytochalasin D (*Figure 6C*, *Video 7*). The same membrane dissociation dynamics were observed when cells expressing GFP-Lpd$^{850–1250aa}$ mutants, (AAPPP)$_{x6}$ and SH3*, were treated with Cytochalasin D (*Figure 6D,E*). Together, these results suggest that the leading edge localization of GFP-Lpd$^{850–1250aa}$ is regulated, in part, by a direct interaction with free actin filament barbed ends, independent of interactions with Ena/VASP or Abi1/endophilin proteins.

## Lpd can tether VASP to actin filaments

The six VASP-binding 'FPPPP' motifs in the C-terminal region of Lpd do not appear to overlap the cloud of basic residues that mediate actin filament binding (*Figure 2A*). We, therefore, wondered whether Lpd could bind Ena/VASP proteins and filamentous actin simultaneously. To test this idea, we first used TIRF microscopy to demonstrate that several Cy3-labeled VASP constructs—an isolated EVH1 domain and two full length, tetrameric, actin-binding mutants—failed to bind actin filaments (*Figure 7A*). Addition of GFP-Lpd$^{850–1250aa}$, however, caused all of these VASP constructs to associate with the sides of actin filaments (*Figure 7B*). Interaction of the tetrameric Cy3-VASP actin-binding mutant with the monomeric Lpd construct produced higher-order VASP clusters visible on the sides of the filaments (*Figure 7B*, arrowheads). We previously observed this type of protein clustering in vitro when we combined tetrameric VASP and ActA$^{255–392aa}$ (unpublished observations).

Supporting the notion that Lpd-VASP complexes can synergistically bind actin filaments, we find that the presence of both VASP and GFP-LZ-Lpd results in a large amount of actin filament bundling in the presence of 100 mM KCl (*Figure 7—figure supplement 1*). In contrast, individual actin filaments

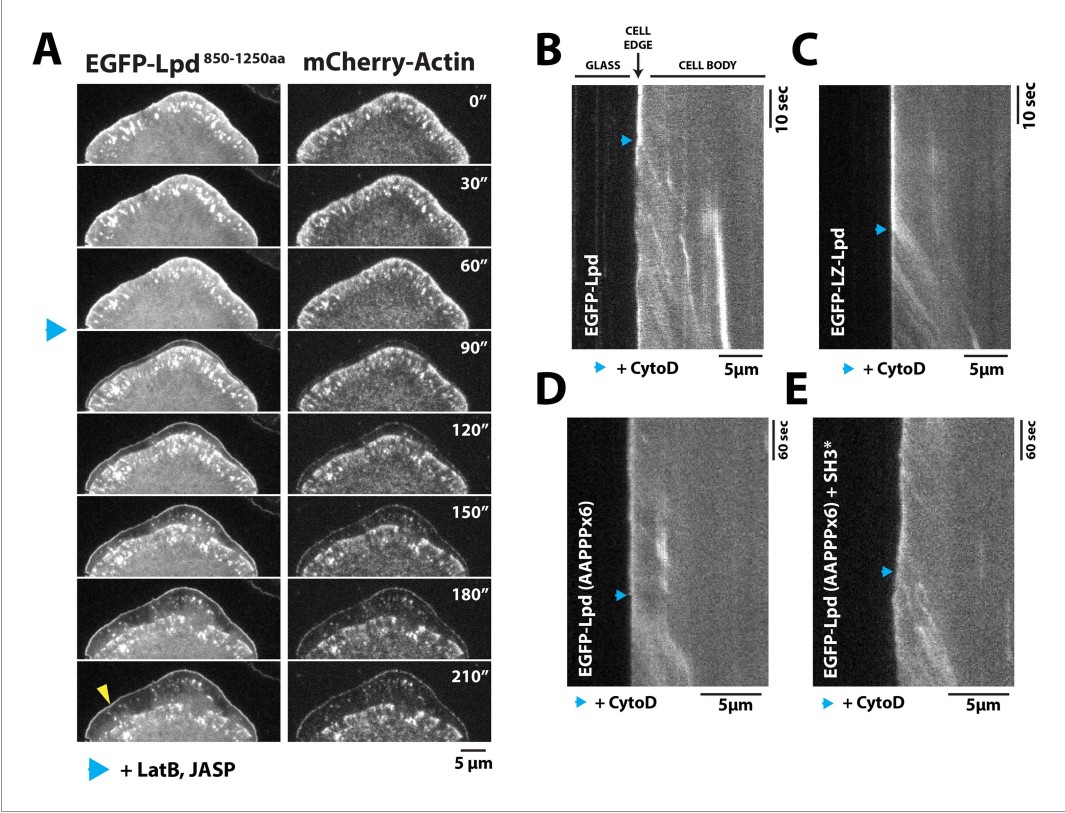

**Figure 6**. Dynamic actin filament assembly and free barbed ends are required for leading localization of GFP-Lpd (850–1250aa). (**A**) Dynamic actin assembly is required for maintenance of GFP-Lpd$^{850–1250aa}$ leading edge localization. Image montage showing translocation of GFP-Lpd$^{850–1250aa}$ and mCherry-Actin toward the cell body, following the addition of 8 µM Jasplakinolide and 10 µM Latrunculin **B**. Note that a population of GFP-Lpd$^{850–1250aa}$ remains associated with the peripheral membrane after addition of Jasp-LatB (yellow arrowhead). Horizontal scale bar, 5 µm. Vertical scale bar, 2 min. (**B–E**) Barbed ends are required for plasma membrane localization of (**B**) GFP-Lpd$^{850–1250aa}$, (**C**) GFP-LZ-Lpd$^{850–1250aa}$, (**D**) GFP-Lpd$^{850–1250aa}$ (AAPPP)$_{x6}$, and (**E**) GFP-Lpd$^{850–1250aa}$ (AAPPPx6 + SH3*). Representative kymographs showing membrane dissociation of GFP-Lpd$^{850–1250aa}$, wild-type and mutants, following the addition of 100 nM Cytochalasin D (blue arrowhead). Horizontal scale bar, 5 µm.

elongating in the presence of 50 nM tetrameric VASP or 250 nM dimeric GFP-LZ-Lpd alone do not bundle actin filaments in high ionic strength buffer. This means that the Lpd-VASP interaction can compensate for the weak actin filament binding observed for the individual proteins in buffer containing 100 mM KCl.

## Lpd enhances VASP barbed end processivity

Because human VASP is a weakly processive actin polymerase and Lpd can interact simultaneously with both VASP tetramers and actin filaments, we hypothesized that Lpd might modulate the processivity of VASP by both clustering tetramers and tethering them to actin filaments. We tested this hypothesis by visualizing actin filament elongation in the presence of either monomeric (Lpd$^{850–1250aa}$) or dimeric (LZ-Lpd$^{850–1250aa}$) GFP-Lpd constructs. The weak interaction between monomeric GFP-Lpd$^{850–1250aa}$ and filamentous actin turns out to be further antagonized by the presence of monomeric actin or profilin-actin (*Figure 8A*). Also, we observed a reduction in the rate of barbed end actin filament elongation in the presence 2 µM profilin-actin and soluble GFP-Lpd$^{850–1250aa}$ or GFP-LZ-Lpd$^{850–1250aa}$ (*Figure 8B*), consistent with Lpd binding to (and partially sequestering) actin monomers. In contrast to monomeric Lpd, dimeric GFP-LZ-Lpd$^{850–1250aa}$ could transiently interact with both the sides and barbed ends of single growing actin filaments (*Figure 8D*, *Video 8*). Although GFP-Lpd and GFP-LZ-Lpd associate more transiently with single actin filaments in the presence of monomeric actin, Lpd binds well enough to

promote filament bundling when present at a near stoichiometric concentration relative to monomeric actin (*Figure 8—figure supplement 1*).

We tried to measure the barbed end dwell time of GFP-LZ-Lpd$^{850-1250aa}$, but the dissociation kinetics and diffusivity of Lpd were too rapid. Using lower concentrations of GFP-LZ-Lpd$^{850-1250aa}$ to visualize single barbed end association events did not improve our resolution. We speculate that the comet-like barbed end localization of GFP-LZ-Lpd$^{850-1250aa}$ (*Figure 8D*) may reflect dimers simultaneously binding to the barbed end and near the barbed end, creating a fluorescent speckle.

Because GFP-Lpd and GFP-LZ-Lpd reduce the rate of actin filament elongation and transiently associate with barbed ends, we tested whether VASP could alleviate the inhibitory effect of Lpd. We measured barbed end elongation rates in the presence of 2 μM actin and 2 μM profilin-actin, as well as buffers of various ionic strengths (*Figure 8B,C*). Because Lpd and VASP synergistically and potently bundle actin filaments in 50 mM KCl, we tried increasing the ionic strength (75–100 mM KCl) to minimize side binding, while favoring barbed end association of the Lpd-VASP complex. However, for all conditions tested, we found that the rate of barbed end elongation in the presence of both Lpd and VASP was intermediate between the rates measured in the presence of either Lpd or VASP alone (*Figure 8B,C*). Interpreting these results remains challenging because multiple competing activities contribute to single filament elongation rates.

Because multiple, competing interactions determine the average barbed end elongation rate in the presence of Lpd and VASP, we returned to imaging single Cy3-VASP tetramers to determine whether Lpd modulates VASP's barbed-end processivity. Fluorescently labeled VASP tetramers accumulate on growing actin filament barbed ends decorated with GFP-LZ-Lpd$^{850-1250aa}$ where they form a stable population ($\tau_1 = 0.58 \pm 0.05$ s (73%), $\tau_2 = 2.3 \pm 0.4$ s (27%); *Figure 8E–G*) that dissociates much more slowly than VASP tetramers observed in the absence of GFP-LZ-Lpd ($\tau_1 = 0.49 \pm 0.03$ s; *Figure 8E–G*).

One challenge we faced in visualizing the colocalization of Lpd and VASP on single actin filament barbed ends was that the high solution protein concentrations required to observe complex formation. Unfortunately, these high concentrations were not compatible with single molecule imaging and produced large protein complexes that were unable to associate with growing barbed ends. To circumvent this problem, we used lower concentrations of his$_{10}$GFP-Lpd (850–1250aa) oligomers purified by size exclusion chromatography (*Figure 1E*) and characterized their effect on VASP barbed end processivity in the single actin filament TIRF assay. Under these conditions we observed complexes containing both VASP and Lpd processively track growing actin filament

**Video 5.** Localization of GFP-Lpd$^{850-1250aa}$ and mCherry-Actin in XTC cells following the addition of 8 μM Jasplakinolide and 10 μM Latrunculin B. Images were acquired with a temporal resolution of 10 s using TIRF-M at 23°C. Drugs were added after 150 s of imaging. Video plays at 10 frames per second. Scale bar, 5 μm.

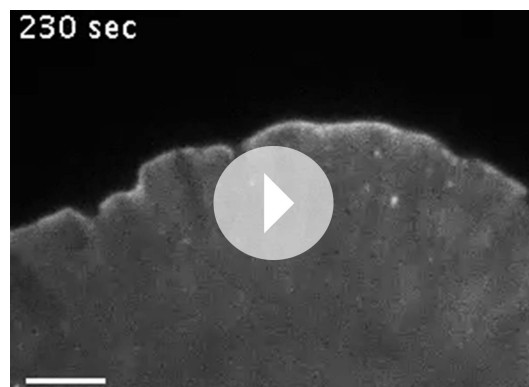

**Video 6.** Localization of GFP-Lpd$^{850-1250aa}$ following the addition of 100 nM Cytochalasin D in a XTC cell imaged with a temporal resolution of 2 s using TIRF-M at 23°C. Cytochalasin D was added at 214 s. Video plays at 50 frames per second. Scale bar, 5 μm.

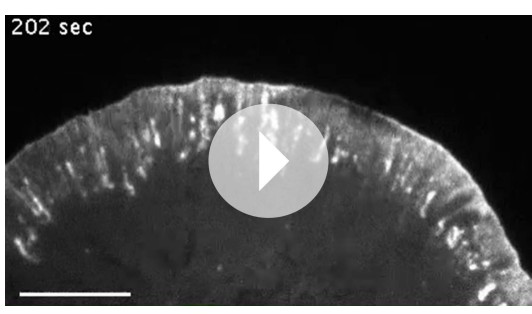

**Video 7.** Localization of GFP-LZ-Lpd$^{850-1250aa}$ following the addition of 100 nM Cytochalasin D in a XTC cell imaged with a temporal resolution of 2 s using TIRF-M at 23°C. Cytochalasin D was added at 102 s. Video plays at 50 frames per second. Scale bar, 5 μm.

barbed ends at a velocity of $36 \pm 9.4$ subunits/s (*Figure 8I*, *Video 9*). The rate at which Lpd-VASP complexes shuttled actin monomers onto free barbed ends was faster than elongation of single actin filaments in the presence of 50 nM tetrameric VASP ($25 \pm 1.6$ subunits/s, *Figure 8I*). The dwell time for VASP-Lpd tip complexes ($\tau_1 = 33 \pm 2$ s; *Figure 8J*) was also significantly longer as compared to single VASP tetramers ($\tau_1 = 0.49 \pm 0.03$ s; *Figure 8F*).

# Discussion

## Lpd binds directly to actin filaments

We discovered a direct interaction between Lpd and filamentous actin that is mediated by a highly basic, unstructured region in the C-terminus of Lpd (residues 850–1250). The cloud of positively-charged amino acid residues is conserved between Lpd homologs of different organisms. These residues, however, are absent from other Ena/VASP binding proteins, such as RIAM, ActA, and Zyxin (*Figure 2B*), suggesting that this basic region is important for cellular functions specific to Lpd. The filament-BD does not overlap motifs that bind the EVH1 domains of VASP (*Figure 2A*), enabling Lpd to simultaneously interact with both Ena/VASP proteins and filamentous actin. Similar to VASP (*Hansen and Mullins, 2010*), monomeric Lpd molecules bind actin filaments only weakly in physiological ionic strength (~150 mM KCl), but bind much more strongly when oligomerized or densely clustered on membranes. Because of this effect, local enrichment of Lpd at the plasma membrane can exert a non-linear, cooperative effect on Ena/VASP activity (*Figure 9*).

Several mechanisms have been proposed to modulate the activity of Ena/VASP proteins, including: post-translational modifications of the EVH2 domain (*Barzik et al., 2005*); alternative splicing of Ena/VASP transcripts (*Philippar et al., 2008*); and specificity for distinct profilin-actin isoforms (*Dugina et al., 2009*; *Mouneimne et al., 2012*). In this study, we establish that Ena/VASP proteins are directly controlled by their membrane-targeting factors. Specifically we show that Lpd-VASP complexes are more persistently localized to polymerizing ends, because the Lpd-actin interaction tethers VASP to the barbed end (*Figure 9*). Filament tethering in combination with VASP clustering (*Breitsprecher et al., 2008*) greatly increases the processivity of Ena/VASP proteins. In addition, Lpd-VASP complexes also enhance barbed-end polymerization more effectively compared to individual VASP tetramers, an ability likely based on Lpd's interaction with actin monomers. By increasing the number of monomer binding sites near the barbed end, Lpd may directly contribute to the polymerase activity of VASP tetramers.

In cells both FL Lpd and its C-terminal actin BD (Lpd$^{850-1250aa}$) localize to actin-based membrane protrusions independently of other binding proteins such as Ena/VASP or Abi1/endophilin. In contrast to FL Lpd, however, we find that acute treatment of cells with cytochalasin D rapidly displaces GFP-Lpd$^{850-1250aa}$ from the leading edge and causes it to translocate toward the cell body with the disassembling lamellipodial actin network. We observed a similar phenotype for Lpd$^{850-1250aa}$ mutants that cannot interact with Ena/VASP proteins or SH3 domains. Taken together these results demonstrate that the C-terminal region of Lpd harbors autonomous filament binding sites that strongly influence its localization to actin rich protrusions. The ability of Lpd$^{850-1250aa}$ to directly interact with polymeric actin extends the canonical model of Lpd membrane localization, which suggests that Lpd targets the plasma membrane by binding to PI(3,4)P$_2$ and/or small GTPases (*Krause et al., 2004*). Future experiments will determine the hierarchical mechanisms that control membrane localization and maintenance of Lpd in response to these diverse molecular inputs.

## Role of profilin in modulating Lpd actin binding

In vitro, the C-terminal actin BD of Lpd reduces barbed end elongation by partially sequestering actin monomers and/or transiently capping free barbed ends. This effect is more pronounced in the presence of profilin-actin, as compared to actin alone, indicating that profilin can shunt Lpd towards the filament barbed end. Interestingly, Lpd contains a poly-proline domain situated upstream of the

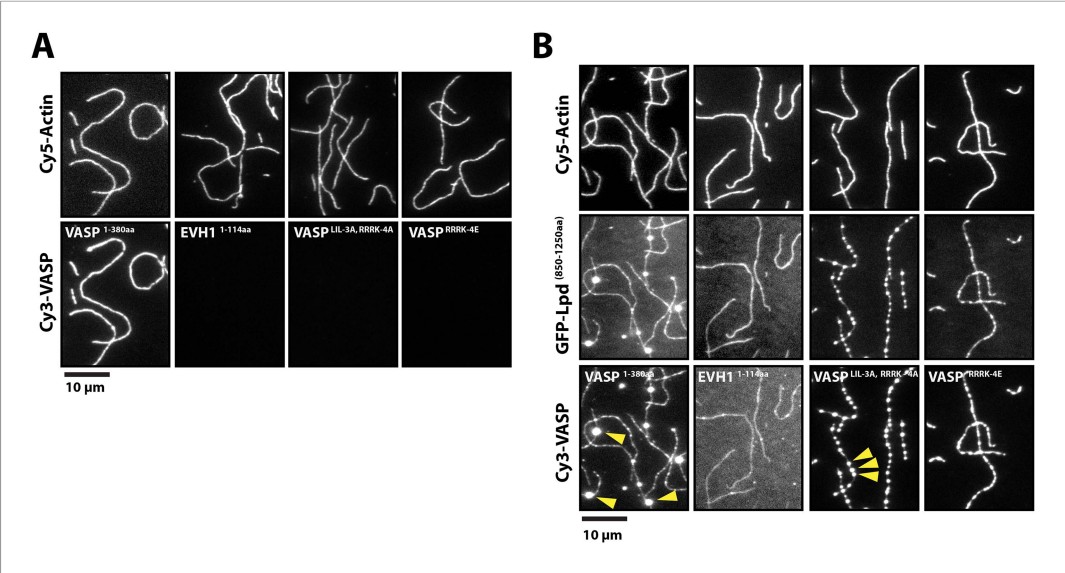

**Figure 7**. Lpd can simultaneously interact with VASP and filamentous actin. (**A**) VASP EVH1 and FL VASP mutants cannot interact with actin filaments in vitro. Images highlight the inability of 200 nM (monomeric concentration) wild-type Cy3-VASP$^{1-380aa}$, Cy3-VASP$^{1-114aa}$ (EVH1 domain), Cy3-VASP$^{LIL-3A, RRRK-4A}$, and Cy3-VASP$^{RRRK-4E}$ to phalloidin stabilized actin filaments (20% Cy5 labeled). Buffer contains 20 mM HEPES [pH 7], 50 mM KCl, 1 mg/ml BSA, 1 mM TCEP. Scale bar, 10 μm. (**B**) GFP-Lpd$^{850-1250aa}$ can simultaneously interact with filamentous actin and VASP EVH1 domains. Colocalization of 500 nM monomeric GFP-Lpd$^{850-1250aa}$ phalloidin stabilized actin filaments (20% Cy5 labeled) in the presence of 200 nM (monomeric concentration) of wild-type Cy3-VASP$^{1-380aa}$, Cy3-VASP$^{1-114aa}$ (EVH1 domain), Cy3-VASP$^{LIL-3A, RRRK-4A}$ (GAB and FAB mutant), or Cy3-VASP$^{RRRK-4E}$ (FAB mutant). Note the formation of large clusters containing VASP and Lpd (yellow arrowheads). Scale bar, 10 μm.

The following figure supplement is available for figure 7:

**Figure supplement 1**. Lpd and VASP synergistically bundle actin filaments.

ABR characterized in this study. We speculate that this poly-proline domain, similar to its function in other actin regulatory proteins such as formins and nucleation promoting factors (*Paul et al., 2008*; *Campellone and Welch, 2010*), could bind profilin-actin and promote its interaction with the actin-binding region, influencing the partitioning of Lpd between monomeric and filamentous actin. Previous work by *Bae et al. (2010)* indicates that the interaction between profilin and phosphatidylinositol lipids (e.g., PI(3,4)P$_2$) plays a key role in regulating Lpd localization to actin-based membrane protrusions. It will, therefore, be important to characterize the interplay between profilin binding to the Lpd poly-proline domain and phosphatidylinositol lipids in the future. We speculate that enrichment of PI(3,4)P$_2$ at the plasma membrane could liberate monomeric actin from profilin-actin complexes, thereby promoting the interaction between Lpd and actin monomers.

## The role of Lpd in actin network assembly

The function of Lpd at the leading edge is not simply to recruit Ena/VASP proteins to the plasma membrane: Knockdown of Lpd inhibits lamellipodium formation more potently compared to loss of Ena/VASP proteins (*Bear et al., 2002*; *Krause et al., 2004*) and results in global reduction of lamellipodial actin density (*Lyulcheva et al., 2008*; *Michael et al., 2010*; *Law et al., 2013*). Recently both Lpd and its binding partners Ena/VASP were found to interact with the Scar/WAVE regulatory complex (*Law et al., 2013*; *Chen et al., 2014*), an important activator of the Arp2/3 complex at the leading edge. These interactions are mediated by the SH3 domain of Abi1, a WAVE complex component, which binds multiple PxxPxR motifs in the C-terminus of Lpd and the EVH1 domains of Ena/VASP proteins. Mutations interfering with WAVE-Lpd binding impaired cell migration in wound closure assays (*Law et al., 2013*), demonstrating the functional relevance of this interaction.

Direct binding of Lpd to filamentous actin might promote Ena/VASP and Scar/WAVE activity within the lamellipodial actin networks they construct. The position of the WAVE-binding motif in Lpd suggests that Lpd could synchronously tether the WAVE regulatory complex and Ena/VASP to filamentous actin. Furthermore, both Lpd (*Krause et al., 2004*; *Chang et al., 2012*) and the WAVE regulatory complex interact with anionic lipids at the plasma membrane (*Lebensohn and Kirschner, 2009*; *Koronakis et al., 2011*). We propose that these multi-layered interactions at the leading edge guide the assembly of the actin nucleation and elongation machinery into supramolecular complexes to coordinate their activities for the efficient construction of force-generating actin networks.

Besides its function in regulating lamellipodial actin network assembly, Lpd has also been implicated in regulating FEME (*Vehlow et al., 2013*; *Boucrot et al., 2015*). Similar to the interactions with the WAVE complex, the C-terminus of Lpd binds to SH3 domains in endophilin using multiple PxxPxR motifs. Consistent with Lpd playing a role FEME, we observed vesicular localization of GFP-Lpd$^{850-1250aa}$ that translocate toward the cell body. Because Lpd interacts directly with filamentous actin and multiple actin regulatory proteins, Lpd could function to orchestrate the actin nucleation machinery at the sites of FEME.

## Materials and methods

### Molecular biology

Lpd clones were derived from a plasmid encoding human Lpd$^{1-1250aa}$ (pBSII-SK(+) vector) provided by Matthias Krause (King's College London). his$_{10}$-TEV-EGFP$^{(A207K)}$-Lpd$^{850-1250aa}$ was cloned into a modified pETM-11 vector (EMBL Protein Expression and Purification Facility; Heidelberg, Germany). Expression vectors used for transient transfection into XTC cell, genes were cloned into the pCMV-EGFP/mCherry (Takara Clontech Laboratories, Mountain View CA, USA) or pDeltaCMV-EGFP vectors originally created by Naoki Watanabe (Tohoku University, Japan) and provided by Orion Weiner (University of California at San Francisco, CVRI). All of our constructs contain a monomeric EGFP variant, A207K point mutation, which significantly reduces the tendency of EGFP to dimerize. Inspired by experiments described by *Inoue et al. (2005)*, we fused a Lyn$_{11}$ palmityolation sequence (GCIKSKGKDSA) derived from Src family kinase to our pDeltaCMV-EGFP(A207K) construct to generate our Lyn-GFP and Lyn-GFP-Lpd fusions. Mutations in the Lpd Ena/VASP BDs, (AAPPP)$_{x6}$, were introduced using Pfu Turbo DNA polymerase and standard site-directed mutagenesis techniques. We made the constitutive GFP-Lpd dimer by inserting a leucine zipper derived from *Saccharomyces cerevisiae* Gcn4p; (*Tomishige et al., 2002*) in between the GFP and Lpd protein sequences [EGFP-GGGY<u>KQLEDKVEELASKNYHLENEVARLKKLVEFGGG</u>-Lpd]. Lpd$^{850-1250aa}$ with all lysine and arginine residues mutated to alanine, called Lpd (44A), was synthesize by Invitrogen (GeneArt) and cloned into the pDeltaCMV-Lyn-EGFP(A207K) or his$_{10}$-TEV-EGFP(A207K) vectors for transient cell expression and bacterial protein expression, respectively. We used the Quikchange Lightning Multi site-directed mutagenesis kit (Agilent, Cat# 210515) to correct lysine/arginine to alanine mutations that flanked or lied within the Abi1/endophilin SH3 domain binding sites resulting in a construct we termed Lpd (35A). This kit was separately used to mutate proline residues within the four predicted SH3 domain-binding sites located in Lpd$^{850-1250aa}$. The pIs for Lpd and homologs were calculated with EXPASY ProtParam (*Wilkins et al., 1999*) and sequences were aligned using ClustalW. Refer to *Figure 5—figure supplement 2* for exact protein sequences of each Lpd$^{850-1250aa}$ mutant used in this study. For a complete list of plasmid DNA used in this study, refer to *Supplementary file 1*.

### Protein expression and purification

BL21 (DE3) Star pRARE bacteria (chloramphenicol resistant) were transformed with his$_{10}$-TEV-EGFP (A207K)-Lpd (850–1250aa, wild-type and mutants) or his$_{6}$-Z-tag-TEV-EGFP(A207K)-LZ-Lpd$^{850-1250aa}$ and plated on LB agar containing 50 µg/ml kanamycin and 34 µg/ml chloramphenicol. The next day, 50–100 ml of TPM media (20 g tryptone, 15 g yeast extract, 8 g NaCl, 2 g Na$_2$HPO$_4$, 1 g KH$_2$PO$_4$, per liter) containing antibiotics was inoculated with multiple bacterial colonies swiped from the agar plate. This starter culture was grown for 4–5 hr at 37°C to an OD600 = 2.5–3.0, before being diluted to OD600 = 0.05–0.1 in 6 liters of TPM media. These cultures were grown to an OD600 = 0.9–1.1 at 37°C. Bacteria were then induced to express Lpd with 0.25 mM IPTG and then shifted from 37°C to 25°C. After 6 hr of expression at 25°C, bacterial cultures were harvested by centrifugation.

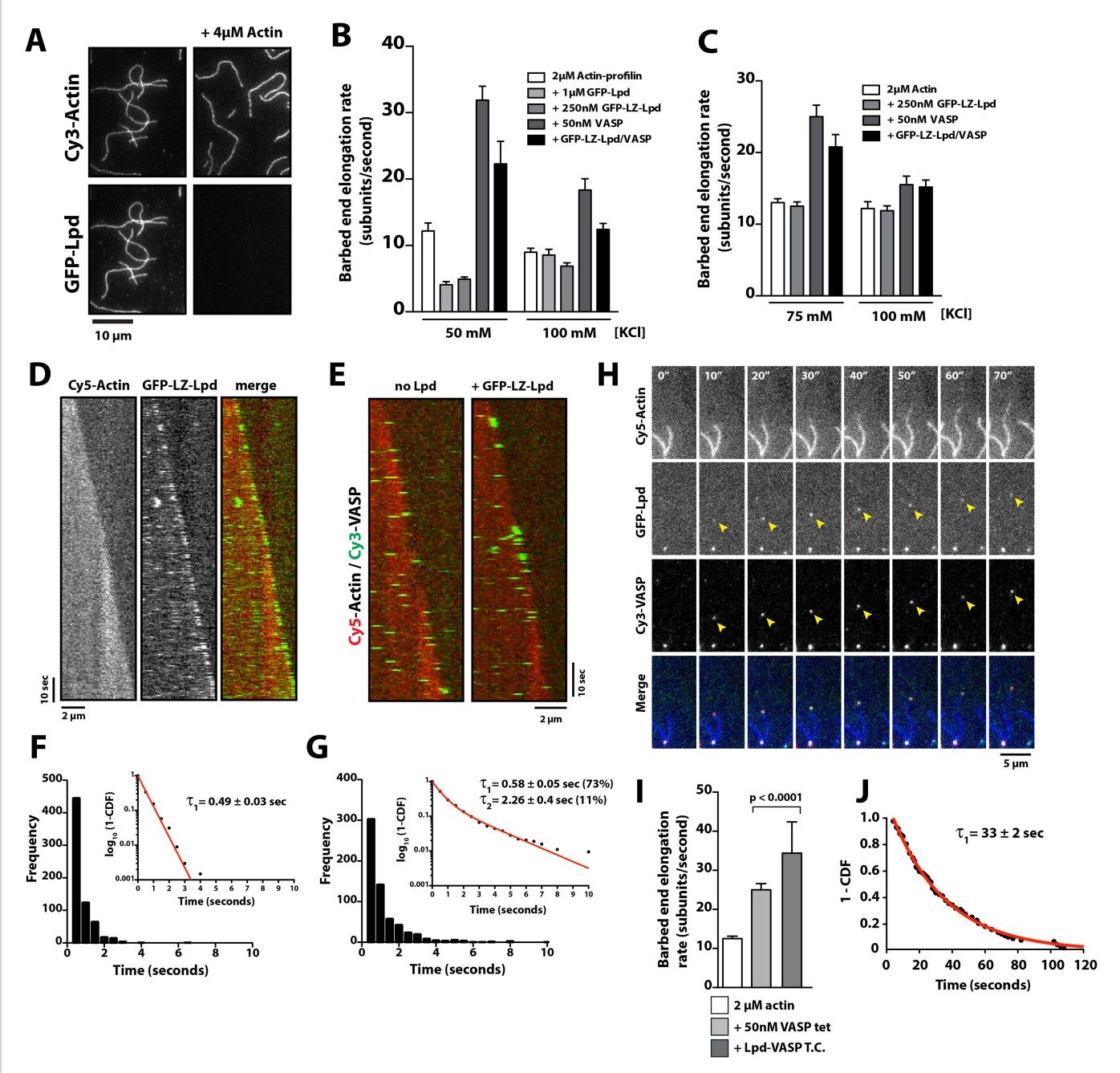

**Figure 8**. Lpd enhances VASP barbed end processivity. (**A**) Monomeric actin antagonizes GFP-Lpd$^{850-1250aa}$ actin filament binding. Visualization of 500 nM GFP-Lpd$^{850-1250aa}$ in the absence or presence of 4 μM monomeric actin in the presence of buffer containing 20 mM HEPES [pH 7.0], 50 mM KCl, 1 mg/ml BSA, 1 mM TCEP, and 25 μM Latrunculin B. Scale bar, 10 μm. (**B**) GFP-Lpd$^{850-1250aa}$ (1 μM, monomer concentration) and GFP-LZ-Lpd$^{850-1250aa}$ (0.25 μM, dimer concentration) slow barbed end elongation in the presence of 2 μM profilin-Mg-ATP-actin (5% Cy5 labeled) and TIRF buffer containing 50 mM KCl. (**C**) Single actin filament elongation rates measured as in (**B**), but in the presence of 2 μM actin (20% Cy5) with TIRF buffer containing 75–100 mM KCl. (**B**, **C**) Error bars represent the standard deviation of the mean (n ≥ 30 barbed end elongation rates measured per condition). (**D**) Dimeric GFP-LZ-Lpd$^{850-1250aa}$ localizes to sides and barbed ends of elongating actin filaments. Kymographs showing the localization of 50 nM GFP-LZ-Lpd$^{850-1250aa}$ (green) to a single actin filament polymerized in the presence of 2 μM Mg-ATP-Actin (20% Cy5, red). Scale bars, 2 μm and 10 s. (**E**) Visualization of processive barbed end associated Cy3-VASP tetramers (green) in the absence or presence of 200 nM GFP-LZ-Lpd$^{850-1250aa}$. Actin filaments were polymerized in the presence of 2 μM Mg-ATP-Actin (20% Cy5, red). Scale bar, 2 μm and 10 s. (**F**, **G**) Calculation of Cy3-VASP barbed end dwell times in the absence (**F**) or presence of 200 nM GFP-LZ-Lpd$^{850-1250aa}$ (**G**) decorated actin filaments. Histogram plots of Cy3-VASP barbed end associated dwell times with insets of the log$_{10}$(1-cumulative distribution frequency) fit with a (**F**) single exponential curve for Cy3-VASP alone ($\tau_1 = 0.49 \pm 0.03$ s, n = 673 molecules) or (**G**) Cy3-VASP

*Figure 8. Continued*

in the presence of 200 nM GFP-LZ-Lpd[850–1250aa] ($\tau_1 = 0.58 \pm 0.05$ s (73%, fast), $\tau_2 = 2.3 \pm 0.4$ s (27%, slow), n = 632 molecules). Note that the dwell times for Cy3-VASP in (**F**) are shorter than previously reported (*Hansen and Mullins, 2010*). This due to Cy5-Actin being a less favorable substrate for barbed end incorporation compared to Alexa488-Actin. (**H**) Clustered $his_{10}$-GFP-Lpd[850–1250aa] increases the processivity of Cy3-VASP. Image montage showing colocalization of the Cy3-VASP (5 nM) and $his_{10}$-GFP-Lpd[850–1250aa] (50 nM) on actin filament barbed end elongating in the presence of 2 μM Actin (20% Cy5) and TIRF buffer contains 75 mM KCl. Note the intensity of the actin filament decreases when the VASP-Lpd complex is associated with the growing actin filament barbed end, indicating that unlabeled vs Cy5-labeled actin is more favorably incorporated. Scale bar, 5 μm. (**I**) Lpd-VASP barbed associated complexes incorporate actin monomers at a faster velocity, as compared to actin filament elongating in the presence of 50 nM tetrameric VASP. Error bars represent the standard deviation of the mean (p-value = $7 \times 10^{-12}$; two-tailed t-test for data sets with unequal variance). (**J**) Calculation of the barbed end dwell times for Cy3-VASP and $his_{10}$-GFP-Lpd[850–1250aa] complexes. Plot of 1-CDF was best fit to a single exponential curve, yielding $\tau_1 = 33 \pm 2$ s (n = 87 complexes).

The following figure supplement is available for figure 8:

**Figure supplement 1**. Lpd dependent actin filament bundling.

Cells were lysed into buffer containing 50 mM $Na_2PO_4$ [pH 8], 400 mM NaCl, 0.4 mM BME (Sigma, Cat# M3148-100ML), 1 mM phenylmethanesulfonyl fluoride (Sigma, Cat# P7626-5G), and DNase (Sigma, Cat# DN25-1G) using a microfluidizer. Lysate was clarified by centrifugation in a Beckman JA-17 rotor for 60 min, 16,000 rpm (35,000 rcf). High speed supernatant was recirculated over a 5 ml HiTrap chelating column (GE Healthcare, Cat# 17-0409-03) that was charged with 100 mM $CoCl_2$ (Sigma, Cat# 255599), washed with water, and then equilibrated with lysis buffer lacking protease inhibitors and DNase. Following capture of his-tagged Lpd, the HiTrap column was washed with ~100 ml of buffer containing 50 mM $Na_2PO_4$ [pH 8.0], 400 mM NaCl, 0.4 mM BME. Proteins were gradient elution over 40 ml (2 ml/min) using a buffer containing 50 mM $Na_2PO_4$ [pH 8.0], 400 mM NaCl, 500 mM imidazole, 0.4 mM BME. Using more than 0.4 mM BME will instantly reduce the chelated $Co^{+2}$ and turn the HiTrap column brown.

Removal the $his_6$-Ztag or $his_{10}$, was achieved by TEV protease digestion of the HiTrap eluate during an overnight dialysis in 4 liters of buffer containing 50 mM $Na_2PO_4$ [pH 8.0], 400 mM NaCl, 0.4 mM BME. After 20–24 hr, the dialysate containing TEV cleaved Lpd was recirculated over a HiTrap chelating column to remove uncleaved protein, $his_6$-TEV protease, and other proteins that non-specifically bound during the first purification step. EGFP-Lpd and EGFP-LZ-Lpd were then

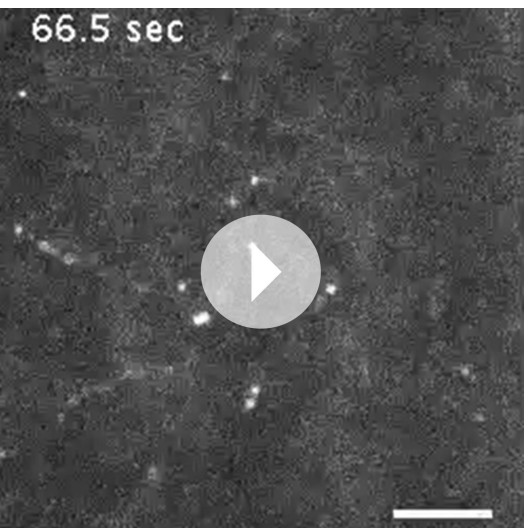

**Video 8.** Dynamic actin filament association of 200 nM dimeric GFP-LZ-Lpd (850–1250aa) in the presence of 2 μM Mg-ATP (20% Cy5 labeled) visualized by TIRF-M. For clarity, only GFP-LZ-Lpd (850–1250aa) is shown. Images were acquired with temporal resolution of 0.5 s using TIRF-M. Video plays at 50 frames per second. Scale bar, 5 μm.

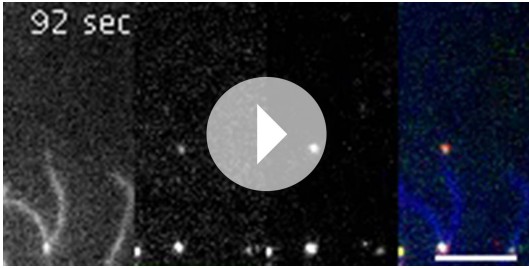

**Video 9.** Visualization of a processive VASP-Lpd tip complex bound to the barbed end of a single actin filament. Elongation of single actin filaments were visualized in the presence of 2 μM Mg-ATP actin (20% Cy5 labeled), 5 nM Cy3-VASP, 50 nM $his_{10}$-GFP-Lpd[850–1250aa], and buffer containing 75 mM KCl. Images were acquired with temporal resolution of 2 s using TIRF-M. Video plays at 20 frames per second. Scale bar, 5 μm.

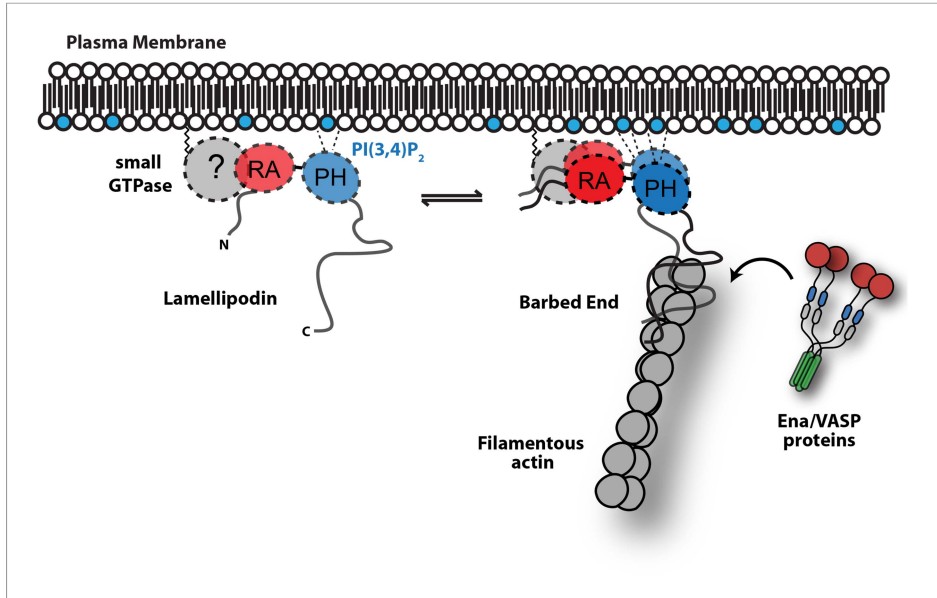

**Figure 9**. Model. Based on the canonical model (*Krause et al., 2004*), Lpd is recruited to actin based membrane protrusions through interactions with phosphatidylinositol lipids (i.e., PI(3,4)P$_2$) and possibly small GTPases (i.e., Ras or Rho family). Similar to the Grb protein family, Lpd is predicted to form homo-dimers mediated by interactions between the coiled-coil and tandem RA-PH domain. We find that the C-terminus of Lpd (residues 850–1250) is sufficient for recruiting Lpd to leading edge membrane where it directly interacts with free barbed ends and/or the sides of the actin filaments. Importantly, this interaction between Lamelliopodin and filamentous actin can occur independently to those mediated by Ena/VASP proteins or SH3 domains (i.e., Abi1/endophilin). However, Ena/VASP proteins recruited to actin based membrane protrusion can simultaneously associate with free actin filament barbed ends and Lpd. By this mechanism, we speculate that the lifetime of membrane targeted and barbed end associated Ena/VASP proteins are extended at the plasma membrane.

buffer exchanged using a HiPrep Desalting 26/10 G25 Sephadex column (GE Healthcare, Cat# 17-5087-01) equilibrated in 20 mM HEPES [pH 7], 75 mM NaCl, 1 mM TCEP. To isolate FL EGFP-Lpd and EGFP-LZ-Lpd, these proteins were loaded onto a MonoS 5/50 GL column (GE Healthcare, Cat# 17-5168-01) equilibrated with 20 mM HEPES [pH 7], 75 mM NaCl, 1 mM TCEP. Bound protein was eluted with a 75–400 mM NaCl salt gradient applied over 40 CV (1 CV = 1 ml) at a flow rate of 1 ml/min. EGFP-Lpd$^{850-1250aa}$ and EGFP-LZ-Lpd$^{850-1250aa}$ eluted in the presence of 170 mM NaCl and 225 mM NaCl, respectively. Peak fractions were pooled, concentrated by centrifugation in a Vivispin 6 (30 kDa MWCO) column. EGFP-Lpd are EGFP-LZ-Lpd were then gel filtered using a 24 ml Superdex200 column equilibrated with 20 mM HEPES [pH 7.5], 200 mM NaCl, 10% glycerol, and 1 mM TCEP. Peak fractions were pooled, concentrated in a new Vivispin 6 (30 kDa MWCO) column, and frozen with liquid nitrogen before being stored in the −80°C freezer.

To isolate uncleaved his$_{10}$-TEV-EGFP(A207K)- Lpd$^{850-1250aa}$, we loaded the HiTrap eluted protein onto a 323 ml Superdex200 gel filtration column equilibrated with 10 mM HEPES [pH 7.0], 200 mM NaCl, 1 mM TCEP. This allowed us to separate soluble aggregates from the monomeric protein. Peak fractions were then combined with 10 mM HEPES [pH 7.5], 1 mM TCEP (1:1 vol) to effectively reduce the salt concentration to 100 mM NaCl. This sample was then loaded on a MonoS column equilibrated with 10 mM HEPES [pH 7.0], 100 mM NaCl, 1 mM TCEP and gradient eluted as described above. Peak fractions of his$_{10}$-EGFP-Lpd eluted from the MonoS column containing a small fraction of spontaneously formed oligomers. Freezing in glycerol, sucrose, arginine-glutamate, or without cryo-protection also caused his$_{10}$-EGFP-Lpd protein to oligomerize. Consequently, his$_{10}$-EGFP-Lpd was frozen after MonoS elution as 300–500 μl aliquots (40–50 μM). When needed for actin bead motility assays and TIRF microscopy experiments, his$_{10}$-EGFP-Lpd was thawed and gel filtered using a 24 ml Superdex200 column equilibrated with 10 mM HEPES [pH 7.5], 200 mM NaCl, and 1 mM TCEP. Freshly gel filtered his$_{10}$-EGFP-Lpd showed no signs to proteolysis or aggregation after at least one week on ice.

Cytoplasmic actin was purified from *Acanthamoeba castellani* as described (*Gordon et al., 1976*; *Hansen et al., 2013*). Actin was stored at 4°C in G-buffer containing 2 mM Tris [pH 8.0], 0.5 mM TCEP, 0.1 mM $CaCl_2$, 0.2 mM ATP, 0.01% azide, and used within 6 months. The quality, or degree of proteolysis, of monomeric actin was routinely monitored by SDS-PAGE. Monomeric actin was labeled on Cys-374 by adding 3–7 molar excess Cy3-maleimide (GE Healthcare, Cat# PA23031) or Cy5-maleimide (GE Healthcare, Cat# PA25031) on ice in G-buffer for 10–15 min. Reactions were quenched with 10 mM DTT and then centrifuged at $346716 \times g$ (TLA 100.4 rotor, Beckman Coulter) to remove insoluble dye and aggregated protein. Labeled actin was then polymerized at room temperature by the addition of KMEI polymerization buffer (final concentration of 10 mM imidazole [pH 7.0], 50 mM KCl, 1 mM $MgCl_2$, 1 mM EGTA, and 1 mM ATP). Labeled filamentous actin was then centrifuged in a TLA 100.4 rotor for 30 min at $195028 \times g$. The pellet was gently washed with G-buffer and then resuspended in G-buffer (2 mM Tris [pH 8.0], 0.5 mM TCEP, 0.1 mM $CaCl_2$, 0.2 mM ATP, 0.01% azide) to initiate actin filament depolymerization. Following depolymerization in G-buffer for 3–5 days, actin was centrifuged at $346716 \times g$ (TLA 100.4 rotor) for 20 min and then gel filtered (Superdex 75) in G-buffer. We typical observed a 50–60% labeling efficiency.

Human VASP was expressed and purified as $his_6$-TEV-KCK-VASP (1–380aa) Cys-light (C7S, C64S, C334A) fusion as previously described by *Hansen and Mullins (2010)*. Mouse EVL was expressed and purified as a $his_6$-TEV-KCK-EVL (1–393aa) Cys-light (C7S, C177S) fusion using the protocol described above. The specific residues mutated in the VASP actin filament binding mutants described in *Figure 4*, are as follows: VASP[LIL-3A, RRRK-4A] (L226A, I230A, L235A, R273A, R274A, R275A, K276A) and VASP[RRRK-4E] (R273E, R274E, R275E, K276E). Both actin binding mutants were expressed and purified in the context of the $his_6$-TEV-KCK-VASP (1–380aa) Cys-light (C7S, C64S, C334A) vector. In all cases the $his_6$ tag was cleaved from purified Ena/VASP protein with TEV protease before cation exchange and size exclusion chromatography (Superdex 200).

The Arp2/3 complex (native, *A. castellani*), capping protein (mouse, recombinant), and profilin (human, recombinant) were purified and handled as previously described by *Akin and Mullins (2008)*. Purification of $his_{10}$-Cherry-SCAR[APWCA] (171-559aa, human WAVE1) was achieved by expressing codon optimized Z-tag-TEV-$his_{10}$-Cherry-SCAR[APWCA] in Rosetta (DE3) bacteria for 20 hrs at 18°C in Terrific Broth media. Cell lysate was recirculated over a 5 ml HiTrap chelating column charged with $CoCl_2$. Eluted protein was cleaved with TEV protease overnight at 4°C and then exchanged into low ionic strength buffer using a HiPrep Desalting 26/10 G25 Sephadex column. FL $his_{10}$-Cherry-SCAR[APWCA] was then separated from partially translated fragments using anion exchange chromatography (i.e., MonoQ) and then separated from protein aggregates by size exclusion chromatography (Superdex75 column).

## Analytical ultracentrifugation

For sedimentation equilibrium experiments, GFP-Lpd[850−1250aa] and VASP[1−114aa] (monomeric EVH1 domain) were diluted into 20 mM HEPES [pH 7], 100 mM KCl, 1 mM TCEP and then centrifuged at $346716 \times g$ (TLA 100.4 rotor, Beckman Coulter) for 20 min to remove proteins that were potentially aggregated. GFP-Lpd[850−1250aa] (9–10 μM) was then combined with 10, 25, 50, 75, or 100 μM VASP[1−114aa] (13.1 kDa) and loaded into 6-well Teflon chambers with quartz windows and placed in a 4 channel Ti-60 rotor. Proteins were centrifuged at 7000, 10,000, and 14,000 rpm in a Beckman XL-I analytical ultracentrifuge at 20°C for 14–18 hr per speed or until equilibrium was reached. Continuous scans of 488 nm absorbance were acquired every 2 hr in replicates of 10, to monitor the sedimentation of GFP-Lpd[850−1250aa] in the absence and presence of VASP[1−114aa]. An extinction coefficient of 55,000 $M^{-1}$ $cm^{-1}$ was used to calculate the GFP-Lpd[850−1250aa] protein concentration from the absorbance measured at 488 nm for each radial position. Global fitting of three equilibrium traces from all speeds (i.e., 7, 10, 14K rpm) for each condition was performed using open source NIH Sedphit and Sedphat software (Peter Schuck, NIH). Equilibrium traces were globally fit using a monomer-dimer self-association model. Using Sednterp, we calculated a partial specific volume of 0.7354 ml/g and a buffer density of 1.00499 g/ml. Our method for fitting the equilibrium traces involved fixing the meniscus position, floating the bottom position, and varying local concentrations in the experimental parameters. Using Sedphat, the apparent molecular weight of GFP-Lpd[850−1250aa] was calculated by fixing the dimer concentration at zero and floating the molecular weight.

## Pegylation of glass coverslips

Glass was functionalized as described by *Bieling et al. (2010)* with some modifications. In brief, coverglass (Corning No. 1.5, 18 × 18 mm sq.) was cleaned by sonication in 3 M NaOH, followed by Pirahna etching (40% hydrogen peroxide, 60% sulfuric acid). Coverslip sandwich were then incubated with undiluted 3-glycidyloxypropyl-trimetoxysilane (GOPTS, Sigma 440167) for 30 min at 75°C. Glass was then washed with anhydrous acetone (Electron Microscopy Sciences; RT 10016) to remove excess silane and rapidly dried with an in house manufactured coverslip spin drier. Next, hydroxyl-$PEG_{3000\ Da}$-$NH_2$ (95%) and $CH_3O$-Biotin-$PEG_{3000\ Da}$-$NH_2$ (5%) powders (Cat# 10-3000-20 and 10-3000-25-20 respectively; Rapp Polymere) and dissolved in anhydrous acetone. Silanized coverslips were assembled into sandwiches with 75 µl of dissolved amino-PEG/amino-PEG-biotin used per coverslip sandwich. The pegylation reaction was performed at 75°C in glass weigh jars (Fisher, Cat# 03-420-5C) for ≥2 hr. Glass was then washed with copious amounts of MilliQ water before spin drying and storing at room temperature in a dust free container.

## Single actin filament TIRF assay

Counterglass slides placed in coplin jars were bath sonicated in 3 M NaOH. Glass was rinsed with water, bath sonicated in 100% ethanol, and then dried. To minimize protein depletion in the TIRF experiments, the counterglass was coated with PLL-PEG (PLL-PEG), which was dried onto the glass surface before rinsing with water. Synthesis of PLL-PEG was performed as previously described by *Huang et al. (2001)*. PEG/biotin-PEG functionalized coverslips were then attached to PLL-PEG coated counterglass with double sided tape (Tesa). To reduce nonspecific binding to the PEG functionalized surface the imaging chamber was washed with a 1× PBS solution pH 7.2 containing 1% Pluronic F-127 and 100 µg/ml beta casein. Glass was then washed with buffer containing 20 mM HEPES pH 7, 1 mM DTT, 200 mM KCl, 100 µg/ml beta-casein (wash buffer). Glass was subsequently incubated for 2 min with 50 nM streptavidin, followed by 50 nM biotin-$PEG_{11}$ heavy meromyosin diluted in wash buffer. To minimize nonspecific binding of GFP-Lpd and actin, it is essential that the stock solution of kappa casein is made fresh in water and centrifuged for 20 min at 346716×*g* (TLA 100.4 rotor or equivalent) to remove aggregated protein. If this is not possible, kappa casein can be left out of the 1% Pluronic F-127 or replaced with beta casein. Ultimately, the assay quality will depend on your ability to block defects on the biotin-PEG functionalized glass surfaces. Because $Lpd^{850-1250aa}$ and human VASP are highly basic proteins, poor coverslip functionalization will result in a tremendous amount of non-specific protein absorption to the glass substrates used for TIRF microscopy.

The single actin filament TIRF assay was performed as described by *Hansen and Mullins (2010)*. When using phalloidin stabilized actin filaments, 2 µM actin (20% Cy3 or Cy5 labeled) was copolymerized with 1 µM dark phalloidin at room temperature for 2–3 hr in buffer containing 10 mM imidazole [pH 7], 50 mM KCl, 1 mM EGTA, and 1 mM $MgCl_2$. For experiments involving dynamic single actin filament elongation, actin polymerization was initiated by combining 1 µl of 10× ME (10× ME contains 0.5 mM $MgCl_2$, 2 mM EGTA) with 9 µl of 4.44 µM monomeric actin (5–20% Cy5 labeled and diluted in G-buffer: 2 mM Tris [pH 8.0], 0.5 mM TCEP, 0.1 mM $CaCl_2$, 0.2 mM ATP, 0.01% azide). Actin was then combined with 2× TIRF imaging buffer (20 µl volume) and 4× protein (i.e., VASP, profilin, Lpd) (10 µl volume) resulting in a final buffer composition of 20 mM HEPES [pH 7], 50–100 mM KCl, 1 mM $MgCl_2$, 1 mM EGTA, 0.2% methylcellulose cP400, 1 mg/ml BSA, 1 mM ATP, 20 mM BME, 1 mM Trolox (Sigma, Cat# 238813), 20 mM glucose, 125 µg/ml glucose oxidase (Serva, #22780.01 *Aspergillus niger*), and 20 µg/ml catalase (Sigma, #C40-100MG Bovine Liver). A final reaction volume of 40 µl (10 µl actin, 20 µl 2× TIRF buffer, 10 µl VASP/Lpd/etc) was flowed through a PEG/biotin-PEG TIRF flow cell, sealed with VALAP (1:1:1 mixture of Vaseline, lanoline, and paraffin wax), and imaged at 23°C.

To calculate the dissociation rate constant ($K_d$) for GFP-$Lpd^{850-1250aa}$ binding to single actin filaments (*Figure 1C*) we used TIRF microscopy to image the density of GFP-$Lpd^{850-1250aa}$ bound to phalloidin stabilized actin filaments (20% Cy5 labeled). For these experiments, increasing concentrations of monomeric GFP-$Lpd^{850-1250aa}$ (0–1 µM) diluted in TIRF buffer containing 50 mM KCl were sequentially flowed into an imaging chamber containing immobilized actin filaments. After a 5 min incubation, GFP-$Lpd^{850-1250aa}$ and Cy5-Actin filaments were imaged across ≥10 fields of view and ≥2 chambers. The average GFP-Lpd fluorescence intensity was calculated across ≥100 µm of filamentous actin.

## Actin cosedimentation assay

Actin polymerization was initiated by mixing 4–20 µM *A. castellani* monomeric actin in buffer containing a final concentration of 20 mM HEPES [pH 7.0], 50 mM KCl, 1 mM MgCl₂, 1 mM EDTA, 1 mM ATP, 1 mM TCEP, in the absence (termed 'native' actin) or presence of an equal molar concentration of dark phalloidin (Calbiochem, Cat# 516640). Actin was then incubated at 23°C for 45–60 min to completely polymerize. In parallel, GFP-Lpd or GFP-LZ-Lpd were diluted in buffer containing a final concentration of 20 mM HEPES [pH 7.0], 50–150 mM KCl (depending on the desired final salt concentration), and 1 mM TCEP. Diluted Lpd (10–20 µM) was then subjected to high-speed ultracentrifugation for 30 min in a TLA100 rotor at 60,000 rpm (156,424×$g$) to remove potentially aggregated protein or particulate matter. Using a spectrophotometer, the Lpd protein concentration was recalculate from the measured absorbance at 280 nm ($\varepsilon_{280}$ = 41,370 M$^{-1}$ cm$^{-1}$ and 44,350 M$^{-1}$ cm$^{-1}$ for GFP-Lpd$^{850-1250aa}$ and GFP-LZ-Lpd$^{850-1250aa}$, respectively). Pre-centrifugation typically results in a loss of 1–2% total protein, which is comparable to the error of most spectrophotometer readings. GFP-Lpd is diluted further to a desired 2× working concentration, which is then mixed 1:1 with 2× pre-polymerized filamentous actin (final reaction volume equals 50 µl). Lpd and filamentous actin are then allowed to equilibrate at 23°C for 60 min. Actin filaments with Lpd bound are then pelleted by ultracentrifugation in a TLA100 rotor at 48,000 rpm (100,111×$g$) for 30 min. After ultracentrifugation, the supernatant is removed and pellets are resuspended in 50 µl of 1× SDS-PAGE protein sample buffer (50 mM Tris-HCl [pH 6.8], 100 mM DTT, 2% SDS, 10% glycerol, 0.1% bromophenol blue). We then load 10 µl of resuspended pellets and load samples that were not centrifuges in a pre-casted NuPAGE 4–12% Bis-Tris gradient gel (Invitrogen, Cat# NP0323BOX), which are then resolved using a Tris/MES/SDS running buffer [pH 7.3]. SDS-PAGE gels were subsequently fixed for 30 min in a solution containing methanol:water:acetic acid (50:43:1), before being stained with Sypro Ruby (Invitrogen, Cat# S12000). Gels are then scanned using a Typhoon imaging system (488 nm/610 nm; EX/EM). The concentration of Lpd and actin in the load and pellet were quantifies by measuring the integrated intensity of individual protein bands in gel after applying a local average background subtraction using ImageQuant. When we process samples containing GFP-Lpd or GFP-LZ-Lpd alone that were included during the co-sedimentation assays (*Figure 1—figure supplement 1D*), we observe the equivalent of 50–100 nM (5–10% of a 1 µM load) Lpd removed from the supernatant in the absence of actin. Because the amount of protein lost during the pre-spin and the co-sedimentation assay are roughly the same, we conclude that protein loss likely reflects non-specific absorption to the walls of the centrifuge tubes. For this reason, we are likely over-estimating the stoichiometry of Lpd bound to actin in *Figure 1I* and *Figure 1—figure supplement 1D* by 5–10%.

## Preparation of SUVs

The following lipids were used to make liposomes: 18:1 (Δ9-Cis) PC (DOPC) 1,2-dioleoyl-*sn*-glycero-3-phosphocholine (Avanti #850375C) and 18:1 DGS-NTA(Ni) [1,2-dioleoyl-*sn*-glycero-3-[(N-(5-amino-1-carboxypentyl)iminodiacetic acid)succinyl] (nickel salt) (Avanti #790404C). All lipids were combined in clean glass vials (National Scientific Glass tubes with PTFE screw cap lid; Fisher 03-391-7B) under a continuous stream of Argon gas. The total amount of lipids used for each liposome preparation was 4 µmoles. Lipid mixtures in chloroform were dried to a film under inert gas, followed by ≥2 hr of drying in a vacuum desiccator. Lipid films were then resuspended with phosphate buffer saline [pH 7.2] to concentration of 4 mM and freeze-thawed 10-times using liquid nitrogen and a water bath at ambient temperature. SUVs were generated by micro-tip sonication (4 × 15 s pulses, 20% amplitude) and then micro-centrifuged at 4°C for 30 min at 21,430×$g$. The supernatant was then transferred to a new eppendorf tube and stored on ice. Vesicles were used the same day.

## Lipid coated glass microspheres

Glass microspheres were washed with nitric acid by diluting a 10% wt/vol slurry of 2.34 µm silica beads (Bangs Laboratories, Cat# SS05N) to 1% wt/vol in a glass vial. Glass microspheres were incubated in nitric acid for 2–3 hr at room temperature. Acid-washed glass beads were pelleted by centrifugation for 5 min and then washed with four times in a glass vial with MilliQ water. Beads were resuspended in water to make a 10% wt/vol slurry.

Lipid bilayers was assembled on acid-washed glass beads by combining 20 µl of 10% bead slurry (vortex/sonicate before aliquoting) with 105 µl 20 mM HEPES [pH 7], 150 mM NaCl (or PBS [pH 7.2]) in

an eppendorf tube. Diluted glass beads were vortexed and then bath sonicated for 5 min. Monodisperse glass beads were then combined with 25 µl of 4 mM SUVs (4 mM = total lipid concentration), vortexed briefly, and then rotated at room temperature for 30 min. After assembling the lipid bilayer, 750 µl of milliQ water was added to each tube and beads were micro-centrifuged for 2 min at 200×$g$. The supernatant was aspirated off the beads and the 750 µl water wash, followed by micro-centrifuged was repeated four times. After the final spin, the supernatant was aspirated leaving ~50 µl of water/beads. Beads were then resuspended by vortexing and 150 µl of buffer containing, 20 mM HEPES [pH 7], 200 mM KCl was added. The final bead slurry was ~1% (wt/vol) and contained a final KCl concentration of 150 mM.

To charge lipid coated glass beads with protein, we combined 5 µl of 1% bead slurry with 45 µl of 111 nM $his_{10}$ tagged protein (i.e., $his_{10}$Cherry-SCAR). Proteins were diluted into buffer containing 20 mM HEPES [pH 7], 150 mM KCl, 100 µg/ml BSA, 0.5 mM TCEP. Before all experiments, we determined the quality of lipid bilayer by charging beads with a mixture of 50 nM $his_{10}$GFP and 50 nM $his_{10}$Cherry-SCAR. If the beads are uniformly coated with GFP and Cherry, you can assume the beads have a continuous and uniform membrane. However, if you observed bright cherry fluorescent patches on glass beads that do not colocalize with GFP, a non-continuous bilayer was generated. We also observed that $his_{10}$Cherry-SCAR, but not $his_{10}$GFP bound to bare glass microspheres under these conditions.

## Actin bead motility assay

The reconstituted actin bead motility mix is based on protocols developed by *Akin and Mullins (2008)*. Our bead motility assay contained 20 mM HEPES pH 7, 100 mM KCl, 1 mM $MgCl_2$, 1 mM EGTA, 1 mM ATP, 0.2% methylcellulose (cP 400), 2.5 mg/ml BSA, 20 mM beta-mercaptoethanol, 7.5 µM cytoplasmic actin (*A. castellani*), 50–125 nM mouse capping protein (recombinant), 100–150 nM Arp2/3 (native, *A. castellani*), 7.5 µM human profilin I (recombinant), 3–5 µM human cofilin (recombinant), and 0.1% (wt/vol) 2.3 µm lipid coated glass beads charged with 100 nM $his_{10}$-Cherry-SCAR[APWCA] for 15–20 min. After reactants were mixed to initiate actin assembly, samples were with quench or immediately flowed in to glass coverslip chamber, which was then sealed with VALAP. In vitro quenching of actin assembly and disassembly was accomplished by combining equal volumes of the bead motility reaction and 37.5 µM Latrunculin B-phalloidin (1:1, therefore 5 molar excess relative to concentration of actin) diluted in buffer containing 20 mM HEPES [pH 7], 100 mM KCl, 100 µg/ml BSA, 0.5 mM TCEP. Imaging chamber for the bead motility assay were assembled with glass silanized with a 2% solution of diethyldichlorosilane (Gelest, Cat# SID 3402.0) in isopropanol (pH 4.5 with acetic acid) (*Akin and Mullins, 2008*).

## Tissue culture and live cell imaging

*Xenopus* fibroblasts (XTC cells) were maintained at 23°C (without $CO_2$) in 70% diluted Leibovitz L-15 media (Invitrogen/Gibco, Cat# 21083-027 no phenol red) containing 10% heat inactivated fetal bovine serum (Gibco) and penicillin/streptomycin (*Watanabe and Mitchison, 2002*). Cells that were 70% confluent in 24-well plastic dishes containing complete media were transiently transfected with 500 ng of plasmid DNA and 1.5 µl Lipofectamine LTX with 1 µl PLUS reagent (Invitrogen, Cat# A12621). After 5 hr of transfection, media was exchanged. Prior to live cell imaging, XTC cells were trypsinized and plated in 70% L-15 media lacking phenol red and FBS. XTC cells were plated onto a glass bottom 96-well plate (Matrical Bioscience, MGB096-1-2-LG) cleaned with 3 M NaOH for 1 hr and subsequently coated with 0.01% (wt/vol) PLL (Sigma, P-8920) for 20–30 min at 23–25°C. Cells were imaged on a Nikon TIRF microscope at 23–25°C.

For experiments in which we froze actin assembly and disassembly with the JLY cocktail, cells were allowed to spread on PLL coated glass for 30 min in the presence of 10 µM Rock kinase inhibitor (Y27632). Cells were then treated with 10 µM Jasplakinolide (CalBiochem, Cat# 420107), 8 µM Latrunculin B (Enzo Life Science, Cat# T110-0001), and 10 µM Y27632 Rock kinase inhibitor (CalBiochem, Cat# 688001) to freeze actin assembly and disassembly in XTC cells (*Peng et al., 2011*). All drugs were diluted to a 2× final concentration in 70% L-15 media lacking serum and penicillin/streptomycin before being added to cells.

B16F1 mouse melanoma cells (ATCC CRL-6323) were cultured in Dulbecco Modified Eagles High glucose containing 10% heat inactivated fetal bovine serum and penicillin/streptomycin. Cells were

grown in 25 cm² flasks at 37°C in the presence of 5% $CO_2$ and split every 3–4 days. For transfection of pCMV-EGFP-Lpd (850–1250aa), cells were plated in 24-well dish to a confluency of 50–60%. We added 0.5–1.0 μg DNA in complex with 2.5–5.0 μl Superfect (Qiagen, Cat# 1006699) to each well containing complete media. After 5–6 hr, the media was changed. 24–36 hr after transfection, cells were prepared for live cell imaging.

Circular glass coverslips (25 mm, Warner Instruments Cat# 64-0715) were cleaned with 3 M NaOH for 30 min at 23°C. Coverslips were rinsed extensively with MilliQ water and coated with 50 μg/ml mouse laminin (Sigma, Cat# 23017-015) diluted in PBS [pH 7.2] at 37°C for 2 hr. Laminin coated coverslips were rinse with PBS and then assembled in stainless steel Attofluor cell imaging chamber (Invitrogen). Transfected cells were then trypsinized in 24-well plastic dishes, centrifuged, wash with complete media, and seeded on the laminin coated glass in the presence of filter sterilized Ham's F12 media containing 10% FBS and 50 mM HEPES [pH 7.2]. Cells spread on laminin coated glass within 30 min. Polarized cells were imaged using wide-field epifluorescence and a Nikon Plan Apo 60× TIRF (NA 1.45) objective on a Nikon Eclipse microscope.

## Microscopy and image processing

XTC cell images were acquired on an inverted Nikon Eclipse TIRF microscope using either a 60× Nikon Plan Apo 60× TIRF (NA 1.45) or a 100× Nikon TIRF (1.49 NA) objective at 23–25°C. B16F1 cells were imaged with Nikon Plan Apo 60× TIRF (NA 1.45) using wide-field epifluorescence illumination at 37°C in the presence of $CO_2$. We employed the Nikon Perfect Focus instrument to maintain the focal plan throughout image acquisition. GFP/Alexa488, Cy3, and Cy5 fluorophores were excited with 491 nm, 561 nm, and 638 nm Coherent lasers respectively. Laser power was modulated with neutral density filters, an AOTF, and exposure settings such that 1–2 mW of laser power was delivered through the objective as measured with a power meter. We used a single filter cube containing a multipass excitation and dichroic filter (Chroma). A rapid switching Sutter Instruments emission filter wheel was positioned before our camera. Images of XTC cells, B16F1 cells, and single actin filament TIRF assay were collected on a cooled Andor Xion EM-CCD camera. The microscope, camera, and lasers were controlled using Micromanager 1.4 (*Edelstein et al., 2010*). Image processing and data analysis was performed using ImageJ. Kymographs were generated using the ImageJ plugin, MultipleKymographs. Data was graphed using Kaleidagraph and Prism. Figures for the manuscript were made in Adobe Illustrator CS6.

## Acknowledgements

We thank Peter Bieling (University of California at San Francisco) for his10-Cherry-SCAR[APWCA] protein and expertise concerning glass surface chemistry; Matthias Krause (King's College London) for the pBSII-SK(+) human Lamellipodin plasmid; Roger Cooke and Kathy Franks-Skiba (University of California at San Francisco) for purified heavy meromyosin; Dave Richmond (Fletcher Lab, UC Berkeley) for liposome preparation protocols; and members of the Mullins, Vale, and Weiner labs for sharing reagents and providing feedback.

## Additional information

### Funding

| Funder | Grant reference | Author |
| --- | --- | --- |
| National Institutes of Health (NIH) | R01, GM061010 | R Dyche Mullins |
| National Science Foundation (NSF) | Graduate Research Fellowship | Scott D Hansen |
| Howard Hughes Medical Institute (HHMI) | | R Dyche Mullins |

The funders had no role in study design, data collection and interpretation, or the decision to submit the work for publication.

### Author contributions

SDH, Conception and design, Acquisition of data, Analysis and interpretation of data, Drafting or revising the article; RDM, Analysis and interpretation of data, Drafting or revising the article

## Additional files

### Supplementary file

• Supplementary file 1. Table of plasmid DNA used for protein expression and cellular transfections.

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
