## [Decision Letter]

Thank you for sending your work entitled “Lamellipodin promotes actin assembly by clustering Ena/VASP proteins and tethering them to actin filaments” for consideration at *eLife*. Your article has been favorably evaluated by Vivek Malhotra (Senior Editor) and three reviewers, one of whom, Pekka Lappalainen, is a member of our Board of Reviewing Editors.

The following individuals responsible for the peer review of your submission have agreed to reveal their identity: Pekka Lappalainen (Reviewing Editor), Alphee Michelot, and Matthias Krause (peer reviewers).

The Reviewing Editor and the other reviewers discussed their comments before they reached this decision. They all found your study interesting and the results of significant importance. However, they also stated that in its current form this work is somewhat too preliminary for publication, and thus additional experiments are needed to confirm the main conclusions and to further strengthen the manuscript. The Reviewing Editor has therefore assembled the following comments to help you prepare a revised submission.

Major concerns:

1) To more reliably and quantitatively examine interaction between Lpd and F-actin, it is best to perform actin filament co-sedimentation assays at physiological ionic conditions for monomeric, dimeric and tri/tetrameric Lpd^850-1250^. These procedures will reveal K_d_ values and stoichiometry of F-actin interactions with Lpd. Ideally, this type of analysis could be also combined with VASP mutants for a more quantitative analysis, in support of the data shown in Figure 6.

2) Interactions of Lpd with filament barbed ends should be analysed further. Does Lpd enhance the ability of VASP to protect barbed ends from capping protein? Furthermore, it would be useful to elucidate whether Lpd interacts with filament barbed ends or binds to the side of actin filament close to the barbed end.

3) Data concerning F-actin dependent localization of Lpd to the leading edge are not particularly convincing, and therefore require additional experiments and revision of the text. It is important to test whether GFP- Lpd^850-1250^ without the palmitoylation signal localizes to the very edge of the cell. Furthermore, from the current data it is not possible to conclude whether Lpd^850-1250^ concentrates to the cell edge by direct interactions with F-actin. This is because Lys/Arg-Ala mutations introduced in this study are also expected to disrupt interactions of Lpd^850-1250^ with SH3 domains (e.g. the ones of endophilin and Abi). Thus, one should either generate additional Lpd mutants, which are not defective in SH3 domain binding, or extensively revise the text to state that the C-terminal region contributes to Lpd localization either through interactions with F-actin or through interactions with other (e.g. SH3 domain containing) proteins.

Reviewer #1 minor comments:

1) VASP also binds palladin, which contributes to stress fiber localization of VASP (Gateva et al., JCS, 2014). This should be mentioned in the Introduction.

2) Because Lpd binds phosphoinositides through its PH domain and is expected to function at the plasma membrane, it is important to test whether the C-terminal region of Lpd can bind actin also in the vicinity of a negatively charged membrane that resembles plasma membrane (because due to its positive charge Lpd C-terminal region itself may associate with negatively charged membranes, which may potentially inhibit F-actin binding). This could be tested e.g. by repeating the assays presented in Figure 1 using beads coated with a more plasma membrane-like lipid composition (e.g. by including 20-40% PE, 10-20% PS and 1-5% PI(4,5)P_2_). At a minimum, it should be stated clearly that experiments in Figure 1 were performed in the absence of negatively charged lipids, and do not necessarily resemble the situation at the plasma membrane.

Reviewer #2 minor comments:

1) “In addition, results from several studies indicate that the MRL proteins likely have additional, Ena/VASP-independent roles in actin network regulation.” This sentence needs references.

2) I would swap Figure 2 and Figure 1. To me, it makes more sense to bring up the simplest system before a more complex one.

3) How do the authors explain the presence of stable His-GFP-Lpd oligomers? The simultaneous presence of monomers and oligomers is generally due to a chemical equilibrium between the monomeric and every oligomeric species. Therefore, I would expect that even after size-exclusion chromatography, the ratio between monomers and oligomers would remain the same. Have the authors checked that point?

4) I did not find the rate of the processive runs in the presence of both VASP and Lpd (Figure 7).

*Reviewer #3 minor comments*:

1)The subheading that membrane tethered Lpd binds actin filaments is an over statement since the data just shows that it slows down actin network assembly mediated by Scar VCA and Arp2/3. It cannot be excluded that it is a direct effect on Scar-VCA-Arp2/3 mediated nucleation.

2) Figure 1: The authors state that membrane tethered Lpd “significantly” slowed the rate of actin network assembly. For this statement they need to provide the statistics and explain what the +/- means: SD or SEM. Looking at the numbers, it is most likely significant.

3) *Xenopus* XTC cells have no real leading edge since they are stuck on poly-L lysine. Live movies in another cell type, such as B16F1 should be performed to confirm the findings for leading edge localization. No need for single molecule imaging here.

4) Figure 7: Was the barbed end polymerization rate measured with GFP-LZ-Lpd and VASP or just VASP on its own in the figure? Comparing GFP-LZ-Lpd and VASP with VASP on its own would give important insights whether GFP-LZ-Lpd blocks barbed ends or works together with VASP to speed up elongation.

5) Figure 7: What are the barbed end dwell times of GFP-LZ-Lpd C-terminal?

[Editors' note: further revisions were requested prior to acceptance, as described below.]

Thank you for resubmitting your work entitled “Lamellipodin promotes actin assembly by clustering Ena/VASP proteins and tethering them to actin filaments” for further consideration at *eLife*. Your revised article has been favorably evaluated by Vivek Malhotra (Senior Editor), a Reviewing Editor, and one of the original reviewers. The manuscript has been improved but there are some remaining, relatively minor, issues concerning actin filament binding assays that need to be addressed before acceptance, as outlined below:

1) Actin filament co-sedimentation assays were performed with a constant concentration of actin and varying the Lpd concentration. However, this is not an optimal way to perform an actin filament co-sedimentation assay, because it does not provide information about how much Lpd sediments in the absence (vs. presence) of actin. Thus, as an additional control experiment, the authors should repeat the assay with a constant, small concentration (e.g. 1 μM) of monomeric and Lpd^850-1250aa^ in the absence and presence of an excess (e.g. 5 μM) of actin.

2) Why do monomeric Lpd and dimeric LZ-Lpd reach the same plateau at saturation of the curves (Figure 1). Does that mean that the two actin binding domains of the LZ-Lpd construct cannot bind simultaneously to F-actin? This should be discussed.

3) Based on the legend to Figure 1—figure supplement 1, the co-sedimentation assays were carried out in the presence of phalloidin. Does Lpd^850-1250aa^ only bind phalloidin stabilized actin filaments (or filaments in the complete absence of actin monomers)? To clearly demonstrate interactions of Lpd^850-1250aa^ with ‘native’ and phalloidin stabilized actin filaments, the co-sedimentation assay described above should be also performed in the presence and absence of phalloidin.

4) The co-sedimentation assay should be described in the Materials and methods section. Furthermore, the authors should clearly explain in the figure legends whether the specific assays in each panel were performed with ‘native’ or phalloidin stabilized actin filaments.

---

## [Author Response]

*1) To more reliably and quantitatively examine interaction between Lpd and F-actin, it is best to perform actin filament co-sedimentation assays at physiological ionic conditions for monomeric, dimeric and tri/tetrameric Lpd*^*850-1250*^*. These procedures will reveal K*_*d*_
*values and stoichiometry of F-actin interactions with Lpd. Ideally, this type of analysis could be also combined with VASP mutants for a more quantitative analysis, in support of the data shown in*
Figure 6.

In response to this comment, we performed additional actin co-sedimentation assays and found that monomeric GFP-Lpd^850-1250aa^ and dimeric GFP-LZ-Lpd^850-1250aa^ bind filamentous actin in buffers containing 50, 100, 150 mM KCl. Consistent with our single-actin filament TIRF measurements, the interaction between actin and Lpd is stronger in low ionic strength buffers. For technical reasons we were unable to achieve sufficiently high concentrations of Lpd to saturate the binding interaction with filamentous actin in 150 mM KCl. Instead, we generated complete binding curves for GFP-Lpd and GFP-LZ-Lpd binding actin in 100 mM KCl. Fitting these curves we obtained dissociation equilibrium constants (K_d_) of 1.49 µM and 3.92 µM for GFP-LZ-Lpd and GFP-Lpd, respectively (Figure 1). Based on ratio of Lpd:Actin sedimented under saturating conditions, we estimate a maximum stoichiometry of one GFP-Lpd^850-1250aa^ to two actin protomers.

We also performed actin co-sedimentation assays in the presence of Lpd and VASP (wild-type and mutants), but these experiments were inconclusive because Lpd and VASP form multi-molecular clusters that pellet independently of their interaction with actin filaments. In place of co-sedimentation, therefore, we used the synergistic bundling of actin filaments by Lpd-VASP to compliment our single-filament Lpd-VASP binding data. We find that individual actin filaments elongating in buffers containing 100 mM KCl and either 50 nM tetrameric VASP or 250 nM dimeric GFP-LZ-Lpd alone do not bundle actin filaments. However, the presence of both VASP and GFP-LZ-Lpd results in a large amount of actin filament bundling. This means that the Lpd-VASP can compensate for the weak actin filament binding observed for the individual proteins in buffer containing 100 mM KCl (Figure 7—figure supplement 1).

*2) Interactions of Lpd with filament barbed ends should be analysed further*. *Does Lpd enhance the ability of VASP to protect barbed ends from capping protein?*

We previously showed that the barbed end association rate constant for capping protein is reduced 6-fold in the presence of 2 µM actin-profilin and 50 nM tetrameric VASP (28). Furthermore, Brietsprecher et al., (2008) showed that clustering enhances VASP anti-capping activity. It is, however, technically challenging to perform an anti-capping experiment in the presence of Lpd because of the prodigious amount of actin bundling that occurs in the presence of both VASP and Lpd. This interaction effectively depletes VASP and Lpd from solution over time and reduces the anti-capping activity of VASP. We, nevertheless, established conditions for measuring capping protein barbed end association in the presence of either Lpd, VASP, or Lpd-VASP in high ionic strength buffer (100 mM KCl). This buffer composition was used to minimize the actin bundling activity of VASP and Lpd, while favoring barbed end association. We find that both Lpd and VASP individually reduce the rate of capping protein barbed end association in the presence of 2 µM actin-profilin. The combined presence of both Lpd and VASP further enhances the anti-capping to a level that is approximately the sum of the individual protein activities. These experiments are difficult to interpret because we cannot quantify how much of the net anti-capping is contributed by VASP alone, Lpd alone, and Lpd-VASP complexes that associate with free barbed ends. Although these results are promising, we think it's premature to include this data in our revised manuscript. If the reviewers have a strong objection, we can try to integrate the anti-capping data into the manuscript.

Ultimately, addressing the physiological anti-capping activity of the Lpd-VASP complex will require characterizing the interaction of these proteins on membranes, in vivo and in vitro. We worked hard to reconstitute Lpd-VASP anti-capping activity using actin networks grown from lipid-coated microspheres. Unfortunately, we have been unable to generate coherent actin networks in the presence of all the proteins necessary to address this question (actin, profilin, his_10_-Cherry-SCAR, his_10_-GFP-Lpd, capping protein, Arp2/3 complex, and VASP).

*Furthermore, it would be useful to elucidate whether Lpd interacts with filament barbed ends or binds to the side of actin filament close to the barbed end*.

To address this comment, it is useful to compare the actin binding activity of GFP-Lpd with that of tropomyosin, as described in our recent study (Hsiao et al., 2015, Current Biology). We found that non-muscle tropomyosin binds the sides of growing actin filaments preferentially *near* the pointed end. In single-filament binding assays we observed many obvious tropomyosin binding events biased toward the pointed end but separated from it by a measurable gap. In contrast, kymograph analysis of GFP-Lpd^850-1250aa^ binding to elongating actin filaments reveals two distinct classes of event: 1) uniform side-binding along the entire filament with no obvious end-bias, and 2) barbed end-binding with no obvious gap between the filament end and GFP-Lpd (Figure 8). We attempted to measure the dwell time of GFP-Lpd^850-1250aa^ on barbed ends with single molecule resolution, but the dissociation and diffusion kinetics are too fast.

In live cells we find that GFP-Lpd^850-1250aa^ rapidly dissociates from the leading edge membrane when barbed ends are blocked by addition of Cytochalasin D. We observe the same effect for both GFP-Lpd^850-1250aa^ wild-type and AAPPP_x6_, indicating that dissociation from the leading edge membrane occurs independently of interactions with Ena/VASP proteins. This result argues that free barbed ends are required for localization of Lpd^850-1250aa^ to the leading edge.

Lpd^850-1250aa^ is unstructured and highly basic, so we propose that a cloud of freely accessible basic residues mediates side-binding to actin filaments. We speculate that, in solution, Lpd is delivered to barbed ends via interaction with monomeric actin or profilin-actin complexes.

Without a high-resolution structure of the Lpd-actin complex we cannot definitively rule out the possibility that Lpd binds preferentially to the sides of filaments near the barbed end. Based on observations described above, however, we are confident Lpd^850-1250aa^ interacts with barbed ends. We have included the above points in the Discussion section of the revised manuscript.

*3) Data concerning F-actin dependent localization of Lpd to the leading edge are not particularly convincing, and therefore require additional experiments and revision of the text. It is important to test whether GFP-Lpd*^*850-1250*^
*without the palmitoylation signal localizes to the very edge of the cell*.

In response to this comment, we tested whether GFP-Lpd^850-1250aa^, lacking the Src palmitoylation sequence, localizes to the leading edge in XTC cells spread on poly-L-lysine. To our surprise, GFP-Lpd^850-1250aa^ localizes to the leading edge even more robustly than Lyn-GFP-Lpd^850-1250aa^. Furthermore, we observed GFP-Lpd^850-1250aa^ undergo retrograde flow (Movie 2) with the same velocity as mCherry-actin speckles in the lamellipodial actin network (Figure 4). Consistent with constitutively dimeric Lpd having a higher affinity for actin, GFP-LZ-Lpd^850-1250aa^ displayed slightly more robust leading edge membrane localization and the intensity of presumptive endocytic particles were brighter compared to those observed in cells expressing GFP-Lpd^850-1250aa^ (Figure 4—figure supplement 1 and Figure 4—figure supplement 2).

*Furthermore, from the current data it is not possible to conclude whether Lpd*^*850-1250*^
*concentrates to the cell edge by direct interactions with F-actin. This is because Lys/Arg-Ala mutations introduced in this study are also expected to disrupt interactions of Lpd*^*850-1250*^
*with SH3 domains (e.g. the ones of endophilin and Abi). Thus, one should either generate additional Lpd mutants, which are not defective in SH3 domain binding, or extensively revise the text to state that the C-terminal region contributes to Lpd localization either through interactions with F-actin or through interactions with other (e.g. SH3 domain containing) proteins*.

To address this comment we constructed a new set of Lpd mutants, designed to more specifically discriminate between different binding partners: Ena/VASP proteins, SH3-binding domains (i.e. Abi1/Endophilin), and actin.

Full-length human Lpd^1-1250aa^ contains 10 potential SH3 domain binding sites (40), while our minimal construct, Lpd^850-1250aa^, contains only four. We mutated essential proline residues in each of these SH3 binding sites to abolish known interactions with Abi1 and endophilin (Figure 5—figure supplement 2). We find that the Lpd SH3* mutant localizes to the leading edge membrane in XTC cells when expressed as a cytoplasmic GFP-Lpd^850-1250aa^ protein or membrane anchored Lyn-GFP-Lpd fusion (Figure 5).

Wild-type -> SH3* Mutations (e.g. PxxPxR -> GxxGxR):

1. KPPPTPQR -> KP**G**PT**G**QR

2. PPPTRPKR -> PP**G**TR**G**KR

3. PTSPK -> **G**TS**G**K

4. RGPPPAPPKR -> RGP**GG**A**GG**KR

Our biochemical experiments indicate that basic residues in Lpd^850-1250aa^ are required for binding filamentous actin. As noted by the reviewers, some of the 44 basic-to-alanine mutations (Lpd 44A) in the construct described in our first manuscript submission fall within SH3 binding sites. To ensure that our charge neutralizing mutations do not disrupt potential interactions with SH3 domain containing proteins, Abi1 and endophilin, we converted 9 alanines in Lpd (44A) back to lysine or arginine (to create Lpd 35A).

44A mutant -> 35A actin binding mutant sequence:

1. AAPPPTPQAN -> A**K**PPPTPQ**R**N

2. APPPTAPAAN -> APPPT**R**P**KR**N

3. VPTSPAS -> VPTSP**K**S

4. AAGPPPAPPAAD -> A**R**GPPPAPP**KR**D

Figure 5—figure supplement 2 contains a complete protein sequence alignment of wild-type and mutant versions of Lpd^850-1250aa^.

Consistent with our original observations, made with Lpd (44A), we find that neutralization of basic residues located between Ena/VASP and SH3 binding sites abolishes leading edge localization. Overall, we find that the localization Lpd^850-1250aa^ is regulated by interactions with a combination of: Ena/VASP proteins, SH3 binding domains (i.e. Abi1/Endophilin), and actin. Based on our live-cell studies, we can rank the effects of mutating various binding sites on leading-edge localization, from most too least deleterious:

1) Loss of basic residues flanking the Ena/VASP and SH3 binding sites

2) Combined SH3* and Ena/VASP binding mutant

3) SH3* binding mutant

4) Ena/VASP binding mutant

In light of these new results, we have replaced most of the Lyn-GFP-Lpd^850-1250aa^ data with new data highlighting the striking localization of the soluble GFP-Lpd^850-1250aa^ construct.

Reviewer #1 minor comments:

1) VASP also binds palladin, which contributes to stress fiber localization of VASP (Gateva et al., JCS, 2014). This should be mentioned in the Introduction.

This was an excellent suggestion. In addition to [25] we also reference [11] in the revised manuscript, as the first paper describing the interaction between palladin and VASP.

*2) Because Lpd binds phosphoinositides through its PH domain and is expected to function at the plasma membrane, it is important to test whether the C-terminal region of Lpd can bind actin also in the vicinity of a negatively charged membrane that resembles plasma membrane (because due to its positive charge Lpd C-terminal region itself may associate with negatively charged membranes, which may potentially inhibit F-actin binding). This could be tested e.g. by repeating the assays presented in*
Figure 1
*using beads coated with a more plasma membrane-like lipid composition (e.g. by including 20-40% PE, 10-20% PS and 1-5% PI(4,5)P2). At a minimum, it should be stated clearly that experiments in*
Figure 1
*were performed in the absence of negatively charged lipids, and do not necessarily resemble the situation at the plasma membrane*.

Additional comments have been added to the main text explaining that the bead motility assay does not included anionic lipids.

Reviewer #2 minor comments:

1) “In addition, results from several studies indicate that the MRL proteins likely have additional, Ena/VASP-independent roles in actin network regulation.” This sentence needs references.

The following references have been added to the main text:

[34]

[44]

[46]

*2) I would swap*
Figure 2
*and*
Figure 1*. To me, it makes more sense to bring up the simplest system before a more complex one*.

As suggested, we have changed the order of the first two figures and updated to the Results section to reflect this change.

3) How do the authors explain the presence of stable His-GFP-Lpd oligomers? The simultaneous presence of monomers and oligomers is generally due to a chemical equilibrium between the monomeric and every oligomeric species. Therefore, I would expect that even after size-exclusion chromatography, the ratio between monomers and oligomers would remain the same. Have the authors checked that point?

The his_10_-GFP-Lpd exists in equilibrium between monomers and oligomers. Dissociation of the oligomers must be slow however, because we can isolate fractions by size exclusion that is enriched in oligomers. In our barbed end processivity experiments (Figure 7), we used 25-50 nM his_10_-GFP-Lpd to form highly processive tip complexes in the presence of 5-10 nM VASP. In these experiments, there also Lpd monomers present which do not contribute to the enhanced processivity of VASP, because they have such a low affinity for filamentous actin.

*4) I did not find the rate of the processive runs in the presence of both VASP and Lpd (*Figure 7*)*.

We measured a rate 34.4 ± 8.0 subunits/sec for Lpd-VASP complexes bound to growing actin filament barbed ends in the presence of 2 µM Actin (20% Cy5). This rate is faster than filaments elongating the presence of 50 nM tetrameric VASP (25 ± 1.6 subunits/sec). These values are now included in the main text and Figure 8.

Reviewer #3 minor comments:

*1) The subheading that membrane tethered Lpd binds actin filaments is an over statement since the data just shows that it slows down actin network assembly mediated by Scar VCA and Arp2/3. It cannot be excluded that it is a direct effect on Scar-VCA-Arp2/3 mediated nucleation*.

We have changed the subheading for this section to the following title: “Membrane-tethered Lamellpodin slows dendritic actin network assembly in vitro”.

*2)*
Figure 1*: The authors state that membrane tethered Lpd “significantly” slowed the rate of actin network assembly. For this statement they need to provide the statistics and explain what the +/- means: SD or SEM. Looking at the numbers, it is most likely significant*.

The +/- values represent the standard deviation. We performed a two tailed t-test for a data set assuming equal variance and calculated a p-value = 3 x 10^-29^. We have added this value to the figure legend and simply stated next to the figure image that p < 0.0001.

*3)* Xenopus *XTC cells have no real leading edge since they are stuck on poly-L lysine. Live movies in another cell type, such as B16F1 should be performed to confirm the findings for leading edge localization. No need for single molecule imaging here*.

We expressed pCMV-EGFP-Lpd^850-1250aa^ in B16F1 cells and found that soluble EGFP-Lpd^850-1250aa^ , lacking the Lyn palmitoyl anchor, strongly localizes to the leading edge membrane of polarized cells migrating on laminin coated glass surfaces (see Figure 4 and Movie 3). Additional text has been added to the Results and Methods sections to describe this result.

*4)*
Figure 7*: Was the barbed end polymerization rate measured with GFP-LZ-Lpd and VASP or just VASP on its own in the figure? Comparing GFP-LZ-Lpd and VASP with VASP on its own would give important insights whether GFP-LZ-Lpd blocks barbed ends or works together with VASP to speed up elongation*.

In our original submission, the single actin filament elongation rates found in Figure 7 were measured in the presence of GFP-Lpd, GFP-LZ-Lpd, or VASP alone with TIRF buffer containing 50 mM KCl. We have now included rates for actin filaments elongating in the presence of both GFP-LZ-Lpd and VASP, which we find to be intermediate to the rates measured in the presence of either VASP or Lpd alone. These results are difficult to interpret for the following reasons:

A) Lpd binds to monomeric actin and profilin-actin, effectively lowering the free actin monomer concentration, which slows actin filament barbed end elongation.

B) Since Lpd appears to interact with the barbed end, meaning Lpd could reduce the association rate constant for actin and profilin-actin by transiently blocking the barbed end.

C) Formation of the Lpd-VASP complex causes both proteins to accumulate on the sides of actin filaments over time. This effectively reduces the concentration of both proteins capable of associating with free barbed ends. As a result, the rate of actin filament elongation is slow in the presence of Lpd-VASP, as compared to VASP alone.

Since many of these effects appear to be dampened by increasing the buffer ionic strength, we performed additional experiments to measure the rate of actin filament elongation in the presence of Lpd, VASP, and Lpd-VASP in high ionic strength buffer.

*5)*
Figure 7*: What are the barbed end dwell times of GFP-LZ-Lpd C-terminal?*

We tried measuring the single-molecule dwell times for barbed end-associated GFP-LZ- Lpd^850-1250aa^. However, the dissociation kinetics and diffusivity along filamentous actin is too fast for us to accurately make these measurements. We speculate that the comet-like barbed end localization of GFP-LZ- Lpd^850-1250aa^ is possibly due to dimers simultaneously binding to the barbed end and near the barbed end, which effectively creates a fluorescent speckle (Figure 8). Please refer to Movie 8 to see how fast GFP-LZ-Lpd binds and dissociates from growing actin filaments.

[Editors' note: further revisions were requested prior to acceptance, as described below.]

*1) Actin filament co-sedimentation assays were performed with a constant concentration of actin and varying the Lpd concentration. However, this is not an optimal way to perform an actin filament co-sedimentation assay, because it does not provide information about how much Lpd sediments in the absence (vs. presence) of actin. Thus, as an additional control experiment, the authors should repeat the assay with a constant, small concentration (e.g. 1 μM) of monomeric and dimeric Lpd*^*850-1250aa*^
*in the absence and presence of an excess (e.g. 5 μM) of actin*.

In response to this comment we performed additional control experiments suggested by the reviewer (see Figure 1—figure supplement 1). The results of these experiments agree with all of our previous observations and further confirm that Lpd does, in fact, bind to actin filaments. Furthermore, this binding does not depend on whether the filaments are stabilized by phalloidin or whether the Lpd is substoichiometric to the actin.

We also carefully investigated the question of how much Lpd aggregates and/or pellets at high speed in the absence of actin. Firstly, frozen GFP-Lpd and GFP-LZ-Lpd remain monomeric and, as judged by analytical gel filtration, does not aggregate after thawing. Secondly, to remove large aggregates that might be present in small numbers we always spin GFP-Lpd and GFP-LZ-Lpd at high-speed prior to combining each protein with filamentous actin (see Methods). Following ultracentrifugation, the protein concentration is then re-measured using a spectrophotometer. This high-speed pre-spin removes less than 1-2% of the total Lpd protein (10-20 µM load), comparable to the error in the spectrophotometer absorbance reading. When we process samples containing GFP-Lpd alone that were included during the co-sedimentation assays (Figure 1—figure supplement 1), we observe the equivalent of 50-100 nM (5-10% of a 1 µM load) Lpd removed from the supernatant in the absence of actin. Because the amount of protein lost during the pre-spin and the co-sedimentation assay are roughly the same, we conclude that protein loss likely reflects non-specific absorption to the walls of the centrifuge tubes. The small amount of Lpd lost to the walls of centrifuge tubes in Figure 1 and Figure 1—figure supplement 1 does not change our overall conclusions. The only significant effect is that the saturating values in Figure 1 would shift down (but not left or right). This does not affect our estimate of the affinity but does alter the apparent binding stoichiometry, which remains difficult to estimate using actin co-sedimentation. To highlight this minor caveat in the revised manuscript, we noted a possible 5-10% over-estimate in the stoichiometry of Lpd bound to actin. We also added a complete description of our co-sedimentation assay to the revised manuscript.

Although we performed the suggested experiment we actually take exception to this comment for several reasons. Firstly, we were asked to perform co-sedimentation experiments to verify the binding data that we measured directly using TIRF microscopy. This appears to be based on the assumption that co-sedimentation is a sort of gold standard for actin filament binding. As I am sure the reviewer is aware, co-sedimentation is an inherently flawed binding assay. It is not an equilibrium method and, therefore, not nearly as accurate as fluorescence-based assays such as polarization anisotropy or direct TIRF imaging as presented in our original manuscript. Co-sedimentation systematically underestimates the affinities of actin binding proteins and fails to detect even physiologically relevant interactions when the dissociation equilibrium constant is too high. This problem is exacerbated when the experiment is performed, as the reviewer suggests, in low (substoichiometric) concentrations of the actin-binding protein. Therefore, while a positive result by co-sedimentation can confirm actin filament binding, a negative result does not provide good evidence against binding.

In addition, it is not fair to say that varying Lpd concentration is “not an optimal way to perform an actin filament co-sedimentation assay.” In fact, this turns out to be a common way of performing the experiment, in part because it is the only way to use co-sedimentation to assess the stoichiometry or cooperativity of actin filament binding.

The only significant virtue of co-sedimentation assays is that they are very easy to perform. Over-reliance on such assays has likely resulted in failure to identify many physiologically significant binding partners of filamentous actin (including Lpd). The concentration of actin filaments in the cortex and leading lamella is extremely high (∼mM) and so to maintain a dynamic cytoskeleton most filament-binding proteins must have fast dissociation rate constants.

*2) Why do monomeric Lpd and dimeric LZ-Lpd reach the same plateau at saturation of the curves (*Figure 1*). Does that mean that the two actin binding domains of the LZ-Lpd construct cannot bind simultaneously to F-actin? This should be discussed*.

The reviewer’s comment suggests that we did not describe this plot well enough in the previous version of our manuscript. For both the x- and y-axes of this plot (Figure 1) we used the concentrations of monomeric GFP-Lpd and GFP-LZ-Lpd, rather than the concentration of dimeric GFP-LZ-Lpd. The concentration of GFP-LZ-Lpd dimer would actually be half the values on the x-axis. The fact that both monomeric and dimeric constructs reach the same plateau at saturation argues strongly that the two binding domains of GFP-LZ-Lpd can simultaneously bind filaments. In contrast, if only one Lpd in the LZ-Lpd construct could interact with actin filaments, we would expect to see a higher concentration of LZ-Lpd in the pellet at the saturation point, as compared to monomeric Lpd under the same conditions. We have rewritten the text and figure caption associated with this experiment to make our results and analysis clearer.

*3) Based on the legend to*
Figure 1—figure supplement 1*, the co-sedimentation assays were carried out in the presence of phalloidin. Does Lpd*^*850-1250aa*^
*only bind phalloidin stabilized actin filaments (or filaments in the complete absence of actin monomers)? To clearly demonstrate interactions of Lpd*^*850-1250aa*^
*with ‘native’ and phalloidin stabilized actin filaments, the co-sedimentation assay described above should be also performed in the presence and absence of phalloidin*.

As noted in the response to the first comment, we performed actin co-sedimentation assays both in the presence and absence of phalloidin (refer to Figure 1—figure supplement 1). The results of these experiments agree with our previous observations that Lpd binds actin filaments regardless of whether they are stabilized by phalloidin. To reinforce this point we have now included TIRF assay data showing that Lpd bundles polymerizing actin filaments in the presence of 2 µM monomeric actin and 1 µM Lpd (refer to Figure 8—figure supplement 1). These results also highlight the fact that Lpd binds ‘native’ actin.

We previously showed that both GFP-Lpd and GFP-LZ-Lpd weakly associate with single actin filaments in the presence of monomeric actin (Figure 8). Based on the reviewer’s comments, this appears to have been interpreted to mean that Lpd does not bind to actin filaments at all in the presence of actin monomers. We made an effort to clarify this point in our newly revised manuscript. We conclude that Lpd weakly associates with single actin filaments in the presence of monomeric actin, but binds well enough to promote filament bundling when present at a near stoichiometric concentration relative to monomeric actin (see Figure 8—figure supplement 1).

*4) The co-sedimentation assay should be described in the Materials and methods section. Furthermore, the authors should clearly explain in the figure legends whether the specific assays in each panel were performed with ‘native’ or phalloidin stabilized actin filaments*.

This point is well taken. We have added a description of the actin co-sedimentation assay to the Materials and methods section of the revised manuscript. We have also revised the figure legends to specify whether ‘native’ or phalloidin-stabilized actin filament were used for each experiment.